

# A model perspective on the dynamics of the shadow zone of the eastern tropical North Atlantic. Part 1: the poleward slope currents along West Africa

Lala Kounta[1,2], Xavier Capet[2], Julien Jouanno[3], Nicolas Kolodziejczyk[4], Bamol Sow[5], and Amadou Thierno Gaye[1]

[1]Laboratoire de Physique de l'Atmosphère et de l'Océan Siméon Fongang, ESP /UCAD, Dakar, Senegal
[2]LOCEAN Laboratory, CNRS-IRD-Sorbonne Universités-UPMC-MNHN, Paris, France
[3]LEGOS Laboratory, IRD-Univ. Paul Sabatier-Observatoire Midi-Pyrénées, Toulouse, France
[4]Laboratoire d'Océanographie Physique et Spatial, IFREMER-IRD-CNRS-UBO, IUEM, Plouzané, France
[5]Laboratoire d'Océanographie, des Sciences de l'Environnement et du Climat, UASZ, Ziguinchor, Senegal

*Correspondence to:* L. Kounta (soxnalala@gmail.com)

**Abstract.**

The West African seaboard is one of the upwelling sectors that has received the least attention and in situ observations relevant to its dynamics are particularly scarce. The current system in this sector is not well known and understood, *e.g.,* in terms of seasonal variability, across-shore structure, forcing processes. This knowledge gap is addressed in a suite of two studies

5   that analyze the mean seasonal cycle of an eddy-permitting numerical simulation of the tropical Atlantic. Part 1 is concerned with the circulation over the West African continental slope at the outmost reach of the Canary current system, between $\sim 10$ and $20^oN$. The focus is on the depth range most directly implicated in the wind-driven circulation (offshore/coastal upwellings and Sverdrup transport), located above the potential density $\sigma_t = 26.7$ kg m$^{-3}$ in the model (approx. above 250 m depth). In this sector and for this depth range, the flow is predominantly poleward as a direct consequence of positive wind stress

10   curl forcing, but the degree to which the magnitude of the upper ocean poleward transport reflects Sverdrup theory varies with latitude. The model poleward flow also exhibits a marked semi-annual cycle with transport maxima in spring and fall. Dynamical rationalizations of these characteristics are offered in terms of wind forcing of coastal trapped waves and Rossby wave dynamics. Remote forcing by seasonal fluctuations of coastal winds in the Guinea Gulf play an instrumental role in the fall intensification of the poleward flow. The spring intensification appears to be related to wind fluctuations taking place at

15   shorter distances, north of the Guinea Gulf entrance and also locally. Rossby wave activity accompanying the semi-annual fluctuations of the poleward flow in the coastal wave guide varies greatly with latitude, which in turn, exerts a major influence on the vertical structure of the poleward flow. Although the realism of the model West African boundary currents is difficult to determine precisely, the present in-depth investigation provides a renewed framework for future observational programs in the region.



# 1 Introduction

The meridional extent of the Canary Current System (CCS) is one of its remarkable features. The northern (resp. southern) extreme is the northern tip of the Iberian peninsula at $\sim 40^{o}$N (resp. Cape Roxo at $\sim 12^{o}$N). Between approximately $35^{o}$N and $20^{o}$N the system is aptly named. The Canary Current is the slow southward return flow of the North Atlantic subtropical gyre

flowing offshore of Morocco (see geographical and oceanographic setting in Fig 1a). About Cape Blanc ($21^{o}$N) the Canary Current bifurcates toward the south-west, away from the African continent and feeds the North Equatorial Current (NEC). Between $\sim20$ and $12^{o}$N, the southern end of the CCS is well separated from the northern CCS (nCCS) by the Cape Verde Frontal Zone (CVFZ) along which flows the NEC. Down to$\sim$200-300 m, major contrasts exist across the CVFZ in terms of water masses (NACW to the north, and fresher SACW to the south). Recent work by Peña-Izquierdo et al. (2015) indicates that

this water mass contrast fades away at greater depth$\sim$300 m, below which northern Atlantic central waters can be found further south. Note that water masses are traditionally separated into surface waters (potential density anomaly $\sigma_t$ lower than 26.3), upper central waters ($26.3<\sigma_t<26.8$) and lower central waters ($26.8<\sigma_t<27.15$) (Elmoussaoui et al., 2005; Rhein and Stramma, 2005; Kirchner et al., 2009; Peña-Izquierdo et al., 2015). South of the CVFZ, the southern Canary Current System will in this study be referred to as Eastern Tropical North Atlantic (ETNA) to underscore its distinct character and avoid overemphasising

oceanic connection and dynamical similarities with the nCCS (to the contrary we will show the importance of the connections with the rest of the tropics).

As we define it the ETNA is further delimited by the West African (WA) shores and, to the south, by the Northern Equatorial Counter Current (NECC) which is surface-intensified and feeds the area with waters of equatorial origin (Richardson and Reverdin, 1987; Blanke et al., 1999). The latitudinal position of the NECC undergoes major seasonal fluctuations as a conse-

quence of the shift in the Intertropical Convergence Zone (ITCZ) and trade wind position (Richardson and Reverdin, 1987; Yang and Joyce, 2006). The NECC northernmost position at$\sim10^{o}$N is reached in late summer/early fall as the NECC flows with maximal intensity. Wind regime exhibits important contrasting specificities in the nCCS and ETNA. From the vicinity of Cape Blanc up to $\sim25$-$30^{o}$N, wind is upwelling favorable all year round. Further north, upwelling winds are increasingly restricted to the summer period. Conversely, the upwelling season is restricted to the winter period, between November and

May, in the ETNA, albeit less so when approaching Cape Blanc. Important contrasts are also found in terms of wind stress curl (WSC; not shown but Sverdrup transport shown in Fig. 1 largely reflects WSC). Except nearshore where coastal wind drop off can be responsible for positive values, WSC is robustly negative over the nCCS (Risien and Chelton, 2008) which belongs to the North Atlantic subtropical gyre. Conversely, ETNA WSC is predominantly positive, in part because the shape of the African continent produces a curvature of the trade winds that favors cyclonic rotation, and also because the ETNA is a

transition region toward the ITCZ (*i.e.,* the trade wind intensity gradually drops southward).

East of $23^{o}$W, the ETNA has historically received limited attention compared to the northern part of the CCS (and other eastern boundary regions) and the regional circulation still suffers from important knowledge gaps. Recently, the issue of the maintainance and possible expansion of the North Atlantic deep oxygen minimum zone has prompted some studies concerned with the density range 26.5-27.2 (Peña-Izquierdo et al., 2015; Brandt et al., 2015; Rosell-Fieschi et al., 2015) within which





low dissolved oxygen concentrations are found. In this study our focus will be on the circulation and dynamics in the ETNA in a distinct slightly lighter density class $\sigma_t \leq 26.7$. This density class, straddling the so-called surface and upper central water ranges, is important because it feeds the coastal upwellings present along Senegal, Mauritania Gambia, and the southern part of Morocco (Glessmer et al., 2009, and Part2). As part of the AWA research program ("ecosystem Approach to the management

of fisheries and the marine environment in West African waters"), we are concerned with the origin of those upwelled waters, the pathways they follow to reach the WA shore, and the dynamics that underlies the existence of these pathways. In addition, note that the relatively low dissolved oxygen concentrations found in this density range have important implications since they contribute to anoxia/hypoxia over the WA continental shelves through coastal upwelling (Brandt et al., 2015; Machu et al., 2017). This element of biogeochemical context is an important motivation to this work.

The ETNA broadly coincides with the shadow zone of the North Atlantic subtropical gyre. In classical wind-driven circulation theories, it is the place of weak circulation owing to the condition of no-flow at the eastern boundary. This means that thermocline waters, including those in our density class of interest, are not directly ventilated (Malanotte-Rizzoli et al., 2000), even though they outcrop overwhelmingly north of $20^o$N in the negative WSC region. Strictly speaking though, the ETNA is not part of the subtropical gyre. As mentioned above, it is characterized by regional positive WSC, positive Ekman pumping

and, in virtue of the Sverdrup relation, northward flow (the vertical distribution of which is not well known and will be an important aspect of the present work). Two main dynamical features have been identified in the region, that are consistent with these expectations, the Guinea thermal dome and the Mauritanian current.

The Guinea dome has been described in numerous studies (Siedler et al., 1992; Stramma et al., 2005) as a permanent quasi-stationary feature on the eastern side of a quasi-zonal thermal ridge present over most of the basin at $\sim$ 12-14$^o$N. The dome is

characterised by a rise of isotherms in the depth range $50 - 300$ m. Voituriez and Herbland (1982) relate the Guinea dome to the cyclonic rotation of the NECC when approaching the eastern end of the basin, from eastward to northward and then westward as the flow connects to the NEC. Their claim is that the thermal crest associated with the NECC is reinforced by this cyclonic rotation, thereby giving rise to the dome structure. Note though that the thermal ridge is much more visible (in meridional cross-sections; Fig. 2a-b) than the thermal dome (in zonal cross-sections; Fig. 2c-d). Despite its supposed importance in conveying

waters rich in dissolved oxygen toward the North-Atlantic OMZ (Peña-Izquierdo et al., 2015), the Guinea dome remains to this date an elusive circulation feature, with limited and contradictory results on its position, dynamics, and variability, as this overview of the literature indicates. Siedler et al. (1992) analyse the Guinea dome structure and seasonal variability using in situ observations and a primitive equation model. Based on temperature distribution and geopotential anomaly fields, they conclude that the Guinea dome is a permanent feature with some seasonal variability, the upper thermocline center of the dome

being found at about $9^o$N, $25^o$W in summer and $10.5^o$N, $22^o$W in winter. Their conclusions partly contradict earlier studies that could not identify the upper thermocline expression of the dome in winter (Voituriez, 1981) and the issue has not been settled since then (Lázaro et al., 2005). Finally, note that ADCP measurements (Stramma and Schott, 1999; Stramma et al., 2005, 2008) or averaged float drifts (Stramma et al., 2008) show weak signs of westward flow on the northern side of the Guinea dome, in contrast to many schematic representations of the circulation associated with the Guinea dome (Stramma and

Schott, 1999; Stramma et al., 2005; Brandt et al., 2010).



Modelling has not led to much clarification, perhaps because the Guinea dome is seldom reproduced with fidelity. The doming of the isopycnals along zonal sections is clearly insufficient in the models of Siedler et al. (1992), Yamagata and Iizuka (1995), as well in ours (see below). In contrast, OFES simulations presented by Doi et al. (2009) tend to overestimate the doming (see their Fig. 2c and d). More problematically, these simulations exhibit ETNA thermohaline uplifts that have distinct

characteristics compared to the real Guinea dome. For instance in Yamagata and Iizuka (1995) the subsurface temperature field exhibits a cold coastal tongue between 10 and $25^o$N that protrudes offshore around $15^o$N (their figure 3). The cold tongue is quasi-stationary while the protrusion is subjected to an important seasonal modulation. Neither the model cold tongue nor the protrusion can unambiguously be identified to match the observed Guinea dome structure.

In terms of dynamical interpretation, the Guinea dome is being systematically related to WSC forcing but two variant

explanations can be found in the litterature: local forcing (Mittelstaedt, 1976; Yamagata and Iizuka, 1995) or large-scale forcing with some Rossby wave effects (Siedler et al., 1992). Historically, this disagreement has flourished in a context of large uncertainties on the WSC patterns (Townsend et al., 2000). The QuikSCAT climatology now allows us to unambiguously demonstrate that the Guinea dome position does not coincide with that of a WSC extremum so the dynamical rationalization for the existence and position of the dome remains to be improved. In Part2 the Guinea dome will be considered in the regional

circulation context. Herein we focus on the flow over the WA continental slope (also referred to as coastal flow given the regional perspective of this work) in the ETNA region, *i.e.,* between approximately 10 and $20^o$N.

In this latitude range, the existence of upper ocean poleward currents over the WA continental slope has been reported for a long time but only the basic aspects of their structure (vertical and horizontal) and seasonal variability are known. Two seasons of intensified poleward flow can be inferred from the litterature (Wooster et al., 1976; Barton, 1989). During the upwelling

season (winter-spring), a poleward undercurrent naturally develops as also found in the other upwelling systems (Hughes and Barton, 1974; Barton, 1998). In summer-fall another poleward flow intensification occurs (Mittelstaedt, 1991). In contrast to the earlier one, the flow is surface intensified and the surface part of the flow is sometimes referred to as the "Mauritanian current" following Kirichek (1971). This characteristic and an approximate coincidence in time has led to suggest that this poleward flow pulse results from the bifurcation of the summer-time northern branch of the NECC as it approaches WA (Kirichek,

1971; Mittelstaedt, 1991; Lázaro et al., 2005). But the specifics of the flow bifurcation (from zonal to meridional) have, to our knowledge, never been described in dynamical terms. Alternatively or complementarily, some authors invoke upwelling wind relaxation south of $\sim 21^o$N (Mittelstaedt, 1991) as the driving process for the summer-fall pulse of poleward flow. Old studies tend to insist on an origin in the Gulf of Guinea for the summer pulse (Kirichek, 1971; Mittelstaedt et al., 1975) but this process has not been revisited for a long time. In his 1989 review study Barton qualifies the knowledge of the poleward

undercurrent along the eastern boundary of the North Atlantic as sketchy: "the arguments for its existence as a continuous entity are based upon relatively few direct current observations, some interpretations of temperature and salinity data, and a degree of speculation". Almost 30 years later, the situation is virtually unchanged with only a few irregular ship ADCP transects to describe the boundary current system (Peña-Izquierdo et al., 2012; Schafstall et al., 2010). In particular, the works of Kirichek (1971); Mittelstaedt (1972); Hughes and Barton (1974) and a few follow-up (Tomczak, 1989) and review studies remain the



main sources of observational knowledge about the WA slope currents between 10 and 20$^o$N (between 5 and 10$^o$N observations
are even fewer).

In this context the point raised by Barton (1989) about whether the poleward undercurrent and the more seasonally inter-
mittent surface countercurrent (aka. the Mauritanian current) are dynamically distinct entities is still pending. The use of two
different names to describe the poleward flow depending on depth range implicitly suggests they are dynamically distinct. We
instead will prefer to use the unique and neutral terminology "WA poleward Boundary Current" (or WABC in short) to refer to
the northward flow present over or in the vicinity of the WA continental slope.

This overview strongly suggests that clarifications are needed on ETNA circulation and dynamics. The present study and
a companion paper (Part 2) are a contribution to this needed effort. The focus is on waters within the density class that most
strongly responds to wind forcing, because they are transported by the Sverdrup flow and/or actively contribute to upwelling
in the ETNA sector, driven by Ekman pumping or coastal divergence. Due to the sparseness of observations in this region
modelling can be an invaluable source of information. Our approach will essentially be based on the careful analysis of an
eddy-permitting NEMO model simulation ((Madec, 2014), see Sec 2). In this Part 1, realism of the modelled circulation and
thermohaline structure are evaluated in Sec. 3 and will be deemed sufficient to inform on several related aspects of the WA
ocean dynamics. The seasonal cycle of the WABC will then be presented (Sec. 4). Its underlying dynamics will subsequently
be explored and discussed, over the continental slope in relation to wind-forced coastal trapped wave theory (Sec.5), and
offshore in relation to Rossby wave theory (Sec. 6). In the light of these results and interpretations, a general assessment of
the knowledge, knowledge gaps, and model biases pertaining to the WA boundary current is proposed (Sec. 7). The source
pathways for WA coastal upwelling waters and the broader regional circulation (which turns out to be of key relevance to
understand these pathways) will be examined in Part 2.

## 2   Data and methods

In this study, we use a numerical model, the oceanic component of the Nucleus for European Modeling of the Ocean program
(NEMO3.6) (Madec, 2014). It solves the three dimensional primitive equations discretized on an Arakawa C grid at fixed
vertical levels (z coordinate). The grid horizontal resolution is 1/4$^o$ and the configuration (refered to as TROP025 hereafter)
generously covers the Tropical Atlantic (35$^o$S–35$^o$N,100$^o$W–15$^o$E). TROP025 has 75 vertical levels, 12 (resp. 24) being con-
centrated in the upper 20 m (resp. 100 m). It is forced at its lateral boundaries with daily outputs from the MERCATOR global
reanalysis GLORYS2V3 (Masina et al., 2015). The open boundary conditions radiate perturbations out of the domain and
relax the model variables to 1 day averages of the global experiment. Details of the method are given in Madec (2014). At the
surface, the atmospheric fluxes of momentum, heat, and freshwater are provided by bulk formulae (Large and Yeager, 2004).
The simulation is forced with the Drakkar Forcing Set DFS5.2 (Dussin et al., 2014) which is mainly based on the ERAinterim
reanalysis (Dee et al., 2011). DFS5.2 consists of 3-hourly fields for wind speed, atmospheric temperature and humidity, and
daily fields for long wave, short wave radiation and precipitation.





Some model-data comparions are available in Da-Allada et al. (2017) and Jouanno et al. (2017). Additional evaluation directly related to this study is presented in the next section. It largely relies on the gridded version of the Coriolis dataset for Re-Analysis version 4.2 (hereafter CORA4.2) for potential temperature and salinity over the period 1990-2014. Its resolution is $0.5^o$ x $0.5^o$ but the gridding software ISAS (Gaillard et al., 2016) relies on correlation length scales that far exceed this mesh grid size, which is detrimental to the representation of boundary currents such as the ones we are interested in. CORA4.2 includes the vast majority of available ARGO profiles and offers a state-of-the art description of the recent thermohaline state of the ocean. The model solution corresponds to a longer period (1979-2015) but this is deemed inconsequential given the limited regional low-frequency variability. In addition, note that a 15-20% fraction of the CORA bins within 1000 km from WA shore have their monthly climatology built with less than 20 T,S vertical profiles (not shown), hence there are large uncertainties in the true ETNA climatological state irrespective of the period. For more information on CORA4.2 readers are refered to Cabanes et al. (2013).

Mathematical symbols have their usual meaning. T, S, $\sigma_t$ and $\rho$ respectively refer to potential temperature, salinity, potential density anomaly, and in situ density. x (resp. y) and u (resp. v) refer to zonal (resp. meridional) directions and velocity. Many diagnostics involve vertical integration between the surface and the isopycnal surface $\sigma_t$ = 26.7 kg m$^{-3}$. U$^{26.7}$ and V$^{26.7}$ are vertically integrated zonal and meridional transports over that depth range. The geostrophic part of the transport is noted with a "g" subscript. At this stage the choice of 26.7 may seem arbitrary but it will be justified by several model analyses below and in Part 2. Mainly, it will be shown that the layer above $\sigma_t$ = 26.7 includes an overwhelming fraction of the Sverdrup and upwelling circulation in the model. $\sigma_t$ = 26.7 is also convenient because subsurface waters lighter than this value are overwhelmingly of the SACW type, in contrast to deeper waters (Peña-Izquierdo et al., 2012, 2015).

For reference, the geostrophic part of the Sverdrup volume transport denoted V$_{sv}$ is defined as (Cushman-Roisin and Beckers, 2011):

$$V_{sv} = \frac{f}{\beta}\left[\partial_x\left(\frac{\tau_y}{\rho_0 f}\right) - \partial_y\left(\frac{\tau_x}{\rho_0 f}\right)\right] \tag{1}$$

where $f$ is the Coriolis parameter, $\beta = \partial_y f$ its derivative with respect to the meridional coordinate, $\rho_0$ is a reference density equal to 1025 kg m$^{-3}$ and $(\tau_x, \tau_y)$ is the surface wind stress (in N m$^{-2}$).

Dynamic heights are presented in Sec. 3. They are calculated at 50 decibar relative to 500 decibar according to the standard formula (Talley, 2011)

$$\Delta D_{50/500} = g\int_{500}^{50}(\rho(T,S,p)^{-1} - \rho(0,35,p)^{-1})\rho_0 dp \tag{2}$$

with the additional assumption that pressure and depth are equivalent. The associated geostrophic flow is classically given by

$$u_g = \frac{g}{\rho_0 f}\int_{500}^{50}\frac{\partial\rho}{\partial y}dz, \qquad v_g = -\frac{g}{\rho_0 f}\int_{500}^{50}\frac{\partial\rho}{\partial x}dz \tag{3}$$

Potential vorticity (PV), which will be examined in the context of Rossby wave dynamics (Sec. 6), is expressed in density coordinates for a layer of fluid of thickness h between the potential density surfaces $\sigma_1$ and $\sigma_2$ (Cushman-Roisin and Beckers,





2011):

$$PV_{\sigma_1}^{\sigma_2} = \frac{f + \xi}{h} \approx \frac{f}{h} \qquad (4)$$

where $\xi$ is the vertical component of relative vorticity for the flow in this fluid layer, which can be neglected given the smallness of the Rossby number associated with the eastern boundary conditions under consideration, $(u, v) \sim 10 \, \text{cm s}^{-1}$ or less.

In several instances, we wish to rotate the flow or wind at the model shelf break to isolate their alongshore/alongslope component. To do so velocities are rotated with respect to the orientation of the shelf break. This orientation is computed at every grid point following the 100 m isobath using centered differences. 15 passes of a 3 point filter with coefficients (0.25,0.5,0.25) are subsequently applied to ensure some smoothness to the angle used for the alongshore projection.

## 3   Model evaluation

In this section we carefully evaluate our simulation with respect to available observations, mainly from the CORA4.2 database. As mentioned in the previous section, time periods for the model and observations do not match precisely. In addition observations are not particularly dense in our region of interest (although many of the bins have $\sim 100$ measurements per climatological month) so we are concerned with a qualitative assessment of model realism.

We start by comparing the monthly climatology of surface zonal currents in TROP025 with the climatology derived from

ARGO drifts obtained by Rosell-Fieschi et al. (2015). The agreement is quite remarkable both in terms of spatio-temporal patterns and current magnitude (compare Fig. 6 with their figure 6). The model captures the northern and equatorial branch of the South Equatorial Current whose separation is most clear in boreal spring, as also found in the observations. More importantly for our study, the NECC seasonal cycle is realistic both in terms of north-south displacement (northernmost extension and widest latitude range in August-September) and change in flow magnitude (swiftest currents >0.3 m s$^{-1}$ found in July-Aug.).

Note though that peak NECC currents in TROP025 seem a bit weaker than observed. The eastward-flowing Guinea current, whose seasonal variability will turn out to be of relevance for the WABC, is also adequately represented. It is intensified between $10^o$W and $0^o$W with a slight peak in boreal summer and a marked decrease in flow speed from September (when the Guinea current is strongest) to November. North of $10^o$N within 5-10$^o$ from the West African coast observed zonal velocity fields are quite noisy but the overall impression is that they tend to be more toward the west than model velocities. This dis-

crepancy may at least partly be related to Stokes drift, which is mainly zonal, can reach about 0.05 m s$^{-1}$ in that region, and affects surfacing ARGO floats (Rosell-Fieschi et al., 2015).

Outside the equatorial region where TROP025 behaves adequately (Da-Allada et al., 2017) we cannot more precisely evaluate the model circulation from direct observations but the equatorial/tropical Atlantic thermohaline structure is overall quite well represented. This is evident from a model-data comparison of climatological temperature along two vertical sections

(Fig. 2) at $13^o$N and $26^o$W. The latter crosses the thermal ridge associated with the NECC. The position of the zonal section is chosen at the latitude typically associated with the Guinea dome. Following Siedler et al. (1992) and Doi et al. (2009) these fields are shown for the Sept.-Oct. period during which the Guinea dome is supposed to be most marked, but model-data



agreement and differences are quite similar for other months. Along $26^oW$ thermocline displacements with latitude are quite faithfully reproduced. For example, the deepening of the $20^oC$ isotherm from $12^oN$ at the top of the thermal ridge to $20^oN$ is $\sim 55$ m in the observations versus 50 m in TROP025. The main bias concerns the sharpness of the thermocline which is insufficient, presumably as a consequence of an overly strong model diapycnal diffusivity. As a consequence, temperature is

$1.5^o$ too high at 100 m depth in the vicinity of the thermal ridge. A similar bias is found along $13^oN$ with a model thermocline that is too diffuse. Along that section, the dome structure is manifest in the model with a deepening of the isotherms colder than $\sim 20^oC$ toward the coast, albeit with less amplitude than found in the observations. As in the observations the longitude east of which the deepening occurs shifts westward for colder isotherms. On the other hand, TROP025 is unable to produce the limited vertical displacement observed east of $20^oW$ for isotherms above $20^o$. So overall, the modelled Guinea dome present

in TROP025 is weaker than observed. This model bias is common (Siedler et al., 1992; Yamagata and Iizuka, 1995), with the notable exception of Doi et al. (2009). We suspect that this bias also reflects in the intensity of poleward velocities which may be underestimated.

    Seasonal climatologies of dynamic height $\Delta D_{50/500}$ (Sec. 2) shown in Fig. 3 for winter and summer confirm this suspicion. Model and observation general patterns are consistent with each other on the NEC and NECC signatures and their winter-

summer changes. The position and intensity of the NECC is quite similar over most of the domain for both seasons, particularly in summer. In winter, the precise form of the cyclonic circulation around the Guinea dome is less well reproduced by the model. The main differences are east of $24^oW$ where the model NECC exhibits some meanders not seen in the observations. A northward branch is evident in the model and observations but the model flow turns more progressively from a more western position at $\sim 25^oW$ than in the observations (*e.g.,* compare geopotential lines 63 and 66 in summer). East of $20^oW$ the tilt

of the geopotential lines is noticeably different in the model and the observations. The regional scale meridional gradient of geopotential in the ETNA is significantly stronger in the observations, respectively 7 versus 4 in summer and 6 versus 2.5 in winter. Implications of this bias will be discussed in Part 2 but we note, again, that model regional circulation is in good qualitative agreement with observations.

    Our analyses will systematically use the isopycnal surfaces $\sigma_t = 26.7$ and, more infrequently, $\sigma_t = 25.2$ because they ap-

proximately limit the range of subsurface waters involved in the model WA coastal upwellings and meridional transport. In Fig. 4 we show the climatological depth of these two isopycnal surfaces in the model and CORA observations . Qualitative agreement between the two is evident, *e.g.,* in terms of outcrop line position, east to west deepening tendency for $\sigma_t = 25.2$ and shape/amplitude of the $\sigma_t = 26.7$ doming in the central part of the basin. Very close to shore along WA the model $\sigma_t = 26.7$ isopycnal surface is not close enough to the surface, presumably because our eddy-permitting resolution is insufficient to ade-

quately resolve the coastal upwelling *per se* (Capet et al., 2008). However this bias is limited to $\sim 20$ m and is not crucial for our continental slope/open ocean investigation.

    In attempting to explain the WABC seasonal cycle we will focus on the dynamics along the coast of the Guinea Gulf. Therefore, a model evaluation is conducted at $4^oN$, $5^oW$, *i.e.,* over the continental slope south of Abidjan, Ivory Coast. The seasonal cycle of temperature T(z) at this location was reported by Picaut (1983) for the period 1957-1964 (see their Fig. 15b).

Comparison with our Fig. 5 reveals the good level of realism of TROP025. Most noticeably, the model produces a semi-

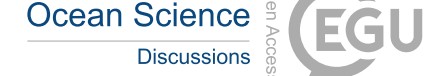



annual oscillation of the thermocline that is most pronounced in the depth range 50-150 m with the highest (resp. lowest) temperatures reached in April and Oct.-Nov (resp. Aug. and Jan.). Model oscillations resemble the observations in terms of phase, upward propagation of the summer-fall doming tendency, and contrast in amplitude between the winter-spring (weak) and summer-fall (strong) oscillations. On the other hand, a quantitative difference concerns the amplitude of the oscillations

which are underestimated by 50% or more in the model, particularly below 150 m. Finally we note that the marked deepening phase between Jul. and Oct. is better captured than the preceding shallowing phase The latter occurs more rapidly in the observations, *e.g.,* over a two months period in June-July for the 16$^o$C isotherm versus 3-4 months in the model. To put these biases into perspective it would be useful to know the degree to which the observations in Picaut (1983) are representative of a long alongshore stretch of ocean, as opposed to very local conditions that TROP025 cannot represent due to lack of resolution.

Overall, the model climatological traits and dominant patterns of seasonal variability are quite realistic, both at basin scale and more locally in the ETNA. Our conclusion is thus that the model circulation and thermohaline structure possesses a sufficient degree of realism to warrant further in-depth analysis. In our discussion of the real ETNA dynamics and circulation (Sec. 7 and likewise in Part 2) we will keep in mind the biases that have also been identified, including the relative weakness of the model poleward flow along WA.

## 15  4   The seasonal cycle of the West African Boundary Current

Poleward boundary currents are ubiquitous along eastern boundary continental slopes (Brink et al., 1983; Huyer, 1983; Connolly et al., 2014) , particularly those subjected to upwelling favorable winds for which subsurface undercurrents are essential flow features (McCreary, 1981; McCreary et al., 1987; Philander and Yoon, 1982). WA is no exception (Barton, 1989). Poleward currents are present in TROP025 as revealed by the seasonally averaged zonal transects of v (Fig. 7). Fig. 8-10 offer

complementary views of the structure and seasonality of the WABC.

At 14$^o$N (*i.e.,* at a central location in the ETNA) a poleward undercurrent is visible over the continental slope for all seasons except in summer (July-Sept.) but it is most marked in fall and to a lesser extent in spring (Fig. 7). (Herafter we refer to these two poleward flow intensification periods as $P_f$ and $P_s$ respectively). The undercurrent appears to be strongly baroclinic with deviations of the isopycnals changing sign in the vertical: upward toward the shore above $\approx$ 75 m depth, and downward below).

Isopycnal displacements reach $\sim$ 100 m for the 26.7 isopycnal between 25$^o$W and 17.5$^o$W. The core of the undercurrent is located at 50 to 100 m depth with peak velocities reaching 6-8 cm s$^{-1}$. In fall the absolute flow maximum is near the surface at $\sim$ 18$^o$W as mixed layer currents are oriented poleward. This surface intensified flow is the model "Mauritanian current". A near-surface secondary maximum is also present in spring at approximately the same longitude. In winter and summer a core of poleward flow present a few hundred kilometers from shore is suggestive of westward propagation of the poleward

undercurrents. This is confirmed with a time-longitude diagram of vertically integrated meridional geostrophic transport $V_g^{26.7}$ about 14$^o$N (integration bounds for the integral follow the isopycnal surface $\sigma_t = 26.7$ and surface, see Sec. 2). The former broadly coincides with the bottom of the poleward flow. The diagram (Fig. 8) exhibits clear signs of westward propagation with speed around 3.5 cm s$^{-1}$ and a dominant wavelength of about 650 km. The signal amplitude decreases dramatically over





3-5$^o$ of longitude. Similar or even shorter scales of attenuation are obtained for other semi-annual Rossby signals emanating from eastern boundary systems (Dewitte et al., 2008; Gómez-Valdivia et al., 2017). The possible reasons underlying this rapid attenuation are discussed in Sec. 6.

As a future point of comparison to other transport estimates, we compute the meridional transport vertically and zonally integrated. Zonal integration is performed from the coast to the first offshore location where the flow changes direction, so the width over which this transport is achieved varies in time. The flow is poleward all year round except for two brief periods of weak equatorward flow in January and July. The poleward transport along the WA boundary is seasonally variable with peak values reaching two Sverdrups or more during the two peak seasons in May-June and Sept.-November, with differences between $P_s$ and $P_f$ being around 20 % (2 Sv in spring vs. 2.5 Sv in fall).

Vertical sections of seasonally averaged alongshore current are shown in Fig. 9. For each latitude, the current intensity is obtained by across-shore averaging the alongshore flow between the 100 m isobath and 150 km offshore (6 grid points), *i.e.,* Fig. 9 is representative of the flow over the continental slope. The regional-scale coherence of the WABC is clearly visible although some minor apparent flow discontinuities result from flow meandering and eddy formation in the vicinity of the major capes as better seen in Fig. 10 (see also Djakouré et al., 2014). The northern bound of the poleward flow varies significantly between $P_s$ and $P_f$ ($\sim$ 20$^o$N versus 25$^o$N respectively). The surface flow is stronger during the latter but poleward currents are otherwise found over a relatively similar depth range that deepens poleward, being located above 100 m (resp. 250 m) depth south of 10$^o$N (resp. between 10$^o$N and 20$^o$N). In fall when the poleward flow reaches further north it extends down to 350-400 m north of 20$^o$N. Note that $\sigma_t = 26.7$ corresponds quite accurately with the bottom depth of the WABC for latitudes between 10 and 20$^o$N, which partly motivated our choice of this isopycnal.

In winter weak but coherent poleward flow is still present over the latitudinal range 7-15 $^o$N. Equatorward flow is mainly found north of 20-22$^o$N in the nCCS where the Canary Current hugs the coast and in the subsurface below the WABC during $P_s$ and $P_f$, which highlights the importance of baroclinic effects in the dynamics of the flow. In winter and summer intense near-surface equatorward flow is found south of $\approx$ 10$^o$N but it is confined to within 50 m from the surface.

The spatio/temporal complexity of the WABC behavior is further revealed in Fig. 10 which shows vertically integrated geostrophic flow from the surface down to $\sigma_t = 26.7$. We make several important observations. First, the general coherence of the flow in the meridional direction (timing of the poleward flow intensifications and their relaxation) is noticeable as well as a general westward propagation tendency. The meridional flow is organized in strips that appear at the coast and move offshore. The strips become increasingly tilted in the northeast/southwest direction as they move offshore, consistent with a faster westward propagation speed closer to the equator. This is particularly evident in March and October when examining the two "phase lines"[1] corresponding to $P_s$ and $P_f$, separated by a thin band of equatorward flow. Propagation becomes increasingly ambiguous when approaching Cape Blanc at about 20$^o$N.

---

[1]We use this terminology in anticipation of an interpretation of this meridional flow signal in terms of Rossby wave dynamics. Nevertheless, a tendency for flow strips to desaggregate is also noticeable, *e.g.,* in July where the strip of poleward flow associated with Ps is broken into several rounded pieces. This underscores the complexity of the dynamics and the possible role of parallel flow instabilities in destabilizing the WABC.





Finally, although this description of the WABC seasonal cycle strongly suggests the importance of its semi-annual cycle, note that a perfect semi-annual oscillation would translate into an exact correspondence between left and right panels in Fig. 10. In contrast, the winter time interval from $P_f$ to $P_s$ appears to be a bit shorter than the summer interval from $P_s$ to $P_f$ (as also confirmed by close inspection of Fig. 8b).

## 5   The WABC coastal dynamics

In the ETNA, positive WSC input is a priori a fundamental ingredient in the generation of poleward flow both nearshore (Fig. 10) as along most eastern boundary systems (Capet et al., 2004; Small et al., 2015), and at larger scale (Sverdrup, 1947, Fig. 1 and see Part 2). To be more quantitative, we compare the theoretical Sverdrup transport $T_{sv}$ and geostrophic WABC transport in TROP025 over the continental slope (Table 1). Geostrophic model transports above 500 m and 1000 m depth as

more classically estimated are also indicated. At 14$^o$N, all model estimates correspond to over 75 % $T_{sv}$ with limited changes when increasing the range of integration. In contrast, model transport increases steadily with the range of integration at 20$^o$, reaching 70 % of $T_{sv}$ when integration goes down to 1000 m.

This being said, there are several reasons why percentages in Table 1) are not strict determinations of the fraction of WABC transport that can be attributed to WSC. First, the cross-shore width and transport of the WABC is not uniquely defined because

it varies as a function of latitude and time of the year (*e.g.,* see our estimation procedure at 14$^o$N used in Fig. 8b). Bottom pressure torque can also cancel part of the WSC contribution to the barotropic vorticity balance (*e.g.,* Molemaker et al., 2015, in the context of an eastern boundary current). In addition, momentum fluxes by mesoscale eddies are known to redistribute WSC input, particularly in the across-shore direction (Marchesiello et al., 2003). Most importantly, the Sverdrup balance is a constraint on the total barotropic flow. Thus, although Sverdrup flow tends to be concentrated in the thermocline and above

(Anderson and Gill, 1975), the WABC transport as we define it (above $\sigma_t = 26.7$) does not solely reflect the Sverdrup balance, but also baroclinic processes and how they vary in time (*e.g.,* on seasonal scales; Figs. 7) and space (*e.g.,* the meridional changes in baroclinic structure; Fig. 9). In this context and pending further progress with model sensitivity experiments we hypothesize that mean poleward transport in the vicinity of the WA continental slope arises from local wind stress curl driving Sverdrup flow plus a combination of baroclinic response to meridional gradient of the Coriolis frequency (Hurlburt and Thompson,

1973), baroclinic response to remote wind forcing in the Gulf of Guinea (McCreary, 1981; Yoon and Philander, 1982) and equatorial band, and alongshore gradient in wind stress curl (Oey, 1999).

We now turn to the seasonal cycle of the WABC about which more can be said based on the TROP025 experiment alone. The main processes underlying $P_s$ and $P_f$ intensified poleward transport could a priori result from four distinct (non mutually exclusive) processes i) local generation of a poleward undercurrent in conjunction with variable coastal upwelling conditions

ii) remote forcing of poleward flow with subsequent propagation in the form of coastal trapped waves iii) local modulation of the nearshore Sverdrup transport in relation with the seasonal cycle of the wind stress curl iv) resonant excitation of free Rossby wave modes at the semi-annual frequency.

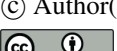



As mentioned above, large deviations from the Sverdrup balance are possible at fine temporal scale (*e.g.,* sub-annual) (Thomas et al., 2015; Wunsch, 2011), particularly near lateral ocean boundaries (Small et al., 2015). One striking discrepancy between the WABC and corresponding Sverdrup transport concerns their respective seasonal cycle which bear no resemblance, as illustrated in Fig. 11 at $14^o$N and $20^o$N. Over the continental slope at $14^o$N, Sverdrup transport is dominated by an annual

cycle that sharply contrasts with the semi-annual cycle of $V_g^{26.7}$ (a similar contrast is found for the geostrophic meridional transport above 500 and 1000 m, not shown). At $20^o$N, the semi-annual cycle of the meridional transport is less prominent but differences with the local Sverdrup transport remain important. These arguments lead us to exclude process iii) as a process responsible for the semi-annual cycle of the WABC. In the remainder of this section the respective roles of i) and ii) are considered while the possible role of iv) will be briefly discussed in the next section.

The significance of semi-annual fluctuations has been established for the circulation of several regions of the equatorial/tropical ocean (Clarke and Liu, 1993). In a nutshell, this arises from having the ITCZ pass twice a year over these regions. The central and eastern Equatorial Atlantic receives significant semi-annual forcing from the winds (Busalacchi and Picaut, 1983). Variability at this frequency is further enhanced by abrupt temporal changes of zonal winds (Philander and Pacanowski, 1986) and basin mode quasi-resonance (Ding et al., 2009; Brandt et al., 2016). The equatorial response can then be propagated

poleward along eastern boundaries by coastal trapped waves, as it occurs in the northern and southern Pacific (Gómez-Valdivia et al., 2017; Ramos et al., 2006) for instance. The Northern Atlantic is peculiar in that coastal trapped waves generated by the reflection of equatorial Kelvin waves in the eastern Gulf of Guinea (GG) have to propagate along a long and corrugated stretch of coastline to reach WA latitudes, with a significant part of the coastline being situated on the edge of the equatorial band at $\sim 4^o$N. This basin geometry is not prone to the coastal transmission of equatorial signals (Philander and Pacanowski,

1986; Polo et al., 2008). Despite some controversies, several studies have dismissed a connection between the equatorial region and the northern GG (Houghton, 1983; Clarke, 1979). To gain further insight into the possible remote sources of semi-annual poleward flow off WA, we computed time-alongshore distance diagrams following the continental slope for the climatological cycle of $18^o$ isotherm depth anomalies ( $z_{18}$, a proxy for thermocline depth anomaly) and $V_g^{26.7}$ anomalies (Fig. 12). The alongshore coordinate covers from $4.2^o$N, $4.75^o$E well into the GG to $25^o$N, $16.5^o$W. A diagram for climatological anomalies

of across-shore Ekman transport is also presented.

Overall, we observe clear signs of long-range poleward propagation for thermocline depth and alongshore flow but this assertion should be qualified given the fact that: 1) propagation is more clearly visible for $z_{18}$ than for $V_g^{26.7}$ (which is considerably noisier) and sea level anomaly (SLA; contours in Fig. 12a) ; 2) propagation is more evident during $P_f$ than $P_s$, with distinct time-space patterns for the two periods; 3) examination of Fig. 12 reveals an off-equatorial maximum in poleward flow

and thermocline depression in the longitude range 4-$14^o$W (*i.e.,* in the GG) with no clear connection to the area east of $0^o$W ; 4) the temporal lag between isothermal depth ($z_{18}$) and geostrophic velocities ($V_g^{26.7}$) seasonal fluctuations is not consistent with Kelvin wave theory; 5) propagation becomes increasingly ambiguous north of Cape Verde at $\approx 15^o$N.





To help with the discussion of 2) and 3) Fig. 13 displays the seasonal cycle of the upwelling index (*i.e.,* Ekman transport) integrated over the stretch of coastline between $3^oE$ and $7.5^oW$:

$$UI_{GG} = \frac{1}{\rho_0 f} \int_{0^oW}^{10^oW} (\tau_x \sin\alpha - \tau_y \cos\alpha)\, ds \tag{5}$$

where $\alpha$ is the angle between the north and the tangent to the coastline leaving land on the right and s is the curvilinear coordinate following the 100 m isobath. $UI_{WAC}$ (resp. $UI_{WA}$) is similarly defined for the longitude (resp. latitude) band $7.5^oW$ and $17^oW$ (resp. $10^o$ and $20^oN$), corresponding to the "West African Corner" between Cape Palmas and Cape Roxo (resp. West African between Cape Roxo and Cape Blanc; see Fig. 1 for locations). All three sectors are of comparable length.

1) is partly expected because alongshore velocities associated with coastal trapped waves shall be approximately geostrophic, hence they depend on the across-shore derivative of $z_{18}$ (Cushman-Roisin and Beckers, 2011). Note also that significant small-scale noise in $V_g^{26.7}$ results from the sinuosity of the flow past topographic irregularities whose position can thus locally depart from the shelfbreak, *e.g.,* in relation to standing meanders in the lee of headlands, as noticeable in Fig. 10 (the position of the main capes is indicated in Fig. 12). The lack of clear propagation tendency found for SLA was previously noticed by Polo et al. (2008) and reflects the fact that sea level over the slope area is largely decoupled from $z_{18}$ (and also from $V_g^{26.7}$). This limits the utility of altimetry to investigate remotely forced dynamics off WA.

Regarding 2) we note that, south of $15^oN$, a propagation phase speed can be identified with reasonable confidence during the $P_f$, especially for $z_{18}$. We estimate c at $\approx 0.9$ m s$^{-1}$ in Fig. 12a and this value is also applicable to $V_g^{26.7}$. This is compatible with low vertical mode coastal trapped wave propagation and, more importantly, consistent with the propagation speed inferred by Picaut (1983) for the GG coastal upwelling signals (0.7-0.8 m s$^{-1}$)[2].

On the other hand, the coherence of the signals expected from poleward propagation is weaker around $P_s$ especially for $V_g^{26.7}$. Even for $z_{18}$ propagation speed is ambiguous and seems to change with time ($\approx 0.2$ m s$^{-1}$ at the transition between negative and positive $z_{18}$ anomalies but a bit faster toward the downwelling peak ($\sim 0.3$ m s$^{-1}$) which roughly coincides with a sign change of $V_g^{26.7}$ north of Cape St Ann (Fig. 12). Such values are untypical for coastal trapped waves. The slow northward shift of the downwelling signal (negative $z_{18}$ and strong poleward flow) may alternatively be attributed to the progressive seasonal migration of the upwelling wind region (related to the seasonal displacement of the ITCZ) but the correspondance between panels a)-b) and c) in Figure Fig. 12 is only partially supportive of this. In addition, propagation at $\approx 0.9$ m s$^{-1}$ may also present, *e.g.,* toward the end of $P_s$ in May. This suggests that both local and remote response to winds combine to produce the winter-spring WABC intensification. Examination of Fig. 13 reveals a complex picture in which each separate coastal sector contributes to upwelling relaxation over a slightly different time period: April to August for WA; March to June for WAC; and March to May for GG. Also note that the GG relaxation is immediately followed by marked increasing upwelling tendency from May to July. Overall forcings over the different sectors largely oppose each other, hence the weakness of propagating oceanic signals and perhaps also the weakness of the poleward flow, relative to $P_f$.

---

[2]Note that our value is $\sim 50$ % slower the one found by Polo et al. (2008) in their numerical simulation over a similar area. Possible reasons for this difference include numerical differences in the grid resolution (higher in TROP025 by a factor 2 and 2.5 in the horizontal and vertical direction respectively) and treatment of viscosity (Hsieh et al., 1983)



With respect to $P_f$ and 3), our analyses suggest the existence of a remote origin for the WABC intensification off WA, with an evident off-equatorial maximum in poleward flow and thermocline depression in the longitude range 4-14$^o$W (*i.e.,* in the GG) in Fig. 12. For the period Oct.-Dec., largest positive values are found in this longitude range. On the other hand, Hovmüller diagrams for $z_{18}$ and $V_g^{26.7}$ exhibit some pattern changes at $\approx$ 0$^o$W near the left edge of Fig. 12. We take this as an indication that the equatorial region is not implicated in the generation of the $P_f$ CTW signal[3]. Examination of 2D monthly regional maps for $z_{18}$ (not shown) confirm the absence of oceanic signal propagation between the equator and the northern part of the GG. Fig. 13 confirms the importance of the GG sector as a forcing region for the poleward flow during $P_f$. An abrupt upwelling relaxation takes place in the GG from August to November when the ITCZ approaches and passes over this sector (Schneider et al., 2014). This relaxation is far steeper than the boreal winter one (compare the two drops in upwelling index $UI_{GG}$ in Fig. 13 and the corresponding local thermal response in Fig. 5 and Picaut 1983, their Fig. 15b). To our knowledge, there has been no mention of the possible role played by the GG wind cycle as a source of remote forcing for the poleward flow in the southern part of the Canary current system (although remote forcing from equatorial origin has been invoked to explain the seasonal cycle of subsurface temperature off Dakar, Busalacchi and Picaut, 1983; McCreary et al., 1984). $UI_{WAC}$ also decreases during $P_f$ so winds in the WAC sector must contribute to WABC intensification but the amplitude of the relaxation is smaller by a factor close to four. The relative importance of the remote forcing associated with each sector depends on their alongshore decay scale, which is poorly constrained and may depend on a number of factors. Limited insight into this question can be gained by comparing $P_f$ and $P_s$ remote forcings.

During $P_s$ the WAC wind relaxation is about twice as intense as during $P_f$ and combines (between April and June) with the relaxation of WA winds. However, the ocean response in terms of poleward flow is significantly weaker than the one during $P_f$ both in terms of current magnitude and meridional extension. GG winds are thus plausibly instrumental in driving the model WABC intensification in fall and, conversely, opposing intensification during most of $P_s$. Further analyses will be needed to clarify this because the seasonal cycle of other environmental parameters may also be involved, *e.g.,* the near-surface density gradient along the waveguide which is larger in spring than in fall (Fig. 9).

With respect to 4), $z_{18}$ and $V_g^{26.7}$ are not precisely in phase as they are expected to be for theoretical Kelvin waves (Cushman-Roisin and Beckers, 2011). A phase shift of the order of one months exists between the two variables with $z_{18}$ lagging, *i.e.,* $P_f$ poleward flow intensification initiates while the thermocline is still in an uplifted position (Fig. 12). A similar discrepancy has been noted before for the California undercurrent and attributed to the effects of Rossby waves dynamics. Due to Rossby waves pressure fluctuations associated with CTWs propagate offshore. In turn, this modulates the alongshore flow which depends on the nearshore-offshore pressure contrast and in particular introduces a phase lead compared to the thermocline depth at the coast Oey (1999). We will discuss the Rossby wave activity offshore of WA in the next section and add support to this explanation.

With respect to 5), propagation of thermocline depth anomalies associated with $P_f$ become progressively elusive beyond Cape Verde[4]. This would not be inconsistent with the major influence exerted by this cape on the poleward flow (Capet et al.,

---

[3]The same is also true for the preceding upwelling phase between June and Sept., in agreement with the conclusions of Clarke (1979) and Houghton (1983). In particular, $z_{18}$ minima reflecting summer upwelling tendency are much more pronounced between 4 and 10$^o$W than near 0-2$^o$E (Fig. 12a).

[4]Likewise, lagged cross correlation of the seasonal 18$^o$C depth between an origin placed 4$^o$E/100m depth and all the other points along the 100 m isobath degrades rapidly north of Cape Verde (not shown).



2017; Alpers et al., 2013) and its dispersive effect on coastal trapped waves (Crépon et al., 1984). The area located between 15 and 20$^o$N (*i.e.,* Cape Verde and Cape Blanc respectively) is also characterized by a rapid shift in the dominant periodicity of $z_{18}$ and $V_g^{26.7}$ fluctuations (Figs. 11 and 12) from semi-annual to annual. In particular, $z_{18}$ variability becomes increasingly complex with reduced magnitude when approaching Cape Blanc where upwelling is permanent. Overall, TROP025 suggests the existence of a transition in this latitude range despite the fact that the WABC can be present up to $\sim 25^o$N in fall.

Overall, no significant forcing at semi-annual period is present north of Cape Palmas (see frequency decomposition of across-shore Ekman transport in Fig. 12) and our analyses indicate that the WABC seasonal cycle is made of two parts that are distinct in terms of forcing mechanism. Strongest poleward flow intensification occurs in fall, both in terms of flow speed and also poleward extension (25$^o$N versus 20-22$^o$N in the model). Such differences seem consistent with existing WABC observations (see Sec. 7) and presumably reflect the strength of remote forcing processes. In fall, poleward flow intensification has been related to a major upwelling wind relaxation in the Gulf of Guinea. In contrast $P_s$ flow intensification appears to be a more complex combination of local and remote responses with time lags resulting in partial compensations.

## 6 WABC and Rossby wave dynamics

As presented above, the across-shore structure of the WABC and its seasonal evolution are strongly suggestive of the important role played by westward Rossby wave propagation. So is the seasonal evolution of the meridional flow patterns that increasingly tilt away from the north-south axis in a clockwise manner as they progress westward (Fig. 10), owing to the rapid change in Rossby wave phase speed in the tropics (Chelton and Schlax, 1996). More precisely, Figs. 7, 8 and 10 are consistent with the presence of a semi-annual Rossby wave coupled to the coastal trapped wave activity. The dispersion relation for linear Rossby waves is reminded here as a starting point (Cushman-Roisin and Beckers, 2011):

$$\omega = -\frac{\beta k}{k^2 + 1/R_n^2} \tag{6}$$

where $R_n$ is the deformation radius for a given vertical mode n which can also be expressed as a function of the gravity wave speed $c_n$ for that mode $R_n = c_n/f$. Dispersion diagrams for different Rossby radii are shown in Fig. 14 including for $R_n$= 50 km which is approximately the value we find at 14$^o$N,18$^o$30$'$W for the first deformation radius, based on the model stratification. Two semi-annual free modes are permitted for this value of $R_n$ but only the one with low wavenumber-long wavelength (k$_2$-$\lambda_2$) has westward energy propagation consistent with our eastern boundary setting. Solving (6) for k gives $\lambda_2$= 731 km, in good agreement with the wavelength estimated from Fig. 8, $\lambda_2^{model}$ = 650 km. Both wavelengths are reported in Fig. 14 as well as the associated theoretical RW phase speeds computed as $c = \omega/k$, which are respectively 4.6 and 4.1 cm s$^{-1}$. This is within 15-30% from the phase speed estimated for the model semi-annual RW (c$_2^{model} \sim$ 3.5 cm s$^{-1}$), again using Fig. 8. In this respect, the propagating signal is thus most consistent with first baroclinic mode RW. But theoretical phase speed for the second baroclinic mode RW is about 2.2-2.5 cm s$^{-1}$ *i.e.,* only 30 to 35 % smaller than the model phase speed.

To elaborate on the relative importance of the different baroclinic modes, an harmonic analysis was performed at each grid cell $(x_i, y_j, z_k)$ over the period 1982-2012 to extract the semi-annual variability of the meridional velocity. The resulting semi-annual harmonics (6 monthly values) were subsequently decomposed onto the baroclinic modes computed for each location



$(x_i, y_j)$ based on a local annual-mean profile of Brunt-Vaisala frequency. Horizontal maps of the depth integrated kinetic energy associated with modes 1-4 are shown in Fig. 16, after time-averaging over the semi-annual cycle. Restricting vertical integration to a layer in which the poleward flow is concentrated, *e.g.,* the upper 200 m, leads to similar results and conclusions. Mode 2 dominates over most of the ETNA except offshore at latitudes beyond $12^oN$ where mode 1 dominates. The shape of the

region having finite values of mode 2 kinetic energy and the general offshore decay are consistent with: energy being mainly radiated from the coastal wave guide; westward energy propagation being more effective at lower latitudes and ineffective north of $12-15^oN$. Similar impression can be drawn for mode 3 and 4 except that westward propagation seems both more strongly damped and more confined meridionally in a low latitude band.

Dominance of mode 2 is a well understood attribute of equatorial/tropical regions (Philander and Pacanowski, 1980). To help

interpret Fig. 16 further, we compute the critical latitudes north of which semi-annual Rossby waves associated with a given mode can no longer freely propagate (Clarke and Shi, 1991). These latitudes correspond to the locations where the discriminant of the quadratic equation for k derived from (6) vanishes. Using the phase speeds indicated in Fig. 15, we find $22^oN$, $11^oN$ and $7^oN$ for the critical latitudes associated with vertical mode 1, 2 and 3 respectively, in agreement with previous estimates (Clarke and Shi, 1991). This is qualitatively consistent with Fig. 16 (*e.g.,* the increasing nearshore confinement of energy with

the mode order) although no sharp transition is found in the modal distribution of energy in TROP025.

Westward propagation of energy away from the coastal guide is an important process contributing to the poleward attenuation of the WABC. Over the continental slope, the progressive deepening of the WABC with increasing latitude in Fig. 9 corresponds to a reduction in the contribution of high-order modes. Although this is consistent with idealized simulations and theoretical arguments (Philander and Yoon, 1982; McCreary, 1981), we cannot be certain that TROP025 motions associated with high-

order modes along WA are more efficiently damped for physical reasons (such as RW generation and frictional processes), as opposed to being dissipated by excessive numerical viscosity/diffusion. For a given mode n dissipation of numerical origin should increase as latitude (resp. the typical horizontal scale associated with that mode $R_n$) increases (resp. decreases). The realism of the model WABC may thus deteriorate with increasing latitude.

In this context, the model behavior in the latitude range $15-22^oN$ requires further clarification. The northern end of this sector

coincides with the critical latitude for baroclinic mode 1 semi annual RWs, hence a potential resonant excitation of these waves because their group velocity vanishes (Hagen, 2005). On the other hand, Hovmüller diagrams similar to Fig. 8 for latitudes between 15 and $22^oN$ reveal a dramatic reduction toward the north of the semi-annual RW signal (not shown but see Figs. 10 and 16 for indirect evidence). This latitude range corresponds to a major transition in the Canary current system in terms of dynamical regime (offshore conditions associated with negative wind stress curl and equatorward flow prevail north of $20^oN$),

two major headlands and abrupt geomorphological near-discontinuities (at Cape Verde and Cape Blanc) and the presence of the permanent Cape Verde thermohaline frontal zone. All these sources of nonlinearities can contribute to the northward weakening of the semi-annual CTW signal and thus prevent the generation of semi-annual RW activity beyond $18-20^o$.

However, the upper ocean potential vorticity (PV, see Sec. 2) field offers the most compelling explanation for the meridional structure of the RW field found in TROP025. Eq. (6) is strictly valid in a large scale ocean at rest in which the only source

of PV gradient is the Coriolis parameter gradient $df/dy = \beta$. In realistic conditions, the large-scale PV field implicated in





the propagation of baroclinic Rossby waves must account for stretching effects associated with background shear flows if
any (Killworth, 1979; de Szoeke and Chelton, 1999). More appropriate quantities to investigate upper ocean RW dynamics
are total PV gradients in three density layers (see definition in Sec. 2): 25.2-26.3 (layer 1), 26.3-26.7 (layer 2) and 26.7-26.9
(layer 3). Layers 1 and 2 are layers in which a large fraction of the WABC transport is concentrated (Fig. 9). They are of
comparable thickness and typically occupy the upper 200-250 m. Layer 3 is also of comparable thickness but it is associated
with a modest fraction of the poleward transport, both nearshore (Fig. 9) and offshore (Fig. 7). PV fields calculated following
(4) are shown in Fig. 17 as well as their gradient vector field. Layer 1 and 2 exhibit relatively similar patterns. PV gradients in
these layers strongly depart from those resulting from variations of the Coriolis parameter alone. In particular, a reversal of the
gradient is found along oblique lines that run North-East to South-West, between Cape Blanc and the Cape Verde islands. RWs
approaching these lines must be subjected to intense dispersive/refractive/dissipative effects. Layer 3 is the deepest layer where
PV gradients are not uniformly oriented toward the north (the gradients vanish in a broad northern sector where stratification
and Coriolis parameter contributions nearly cancel). Below layer 3 PV gradients are dominated by $\beta$ and relatively uniform
from the coast to thousands of kilometers offshore (not shown).

At $14^oN$, the zero PV gradient line in layer 1 (resp. layer 2) is located at $\sim 24^oW$ (resp. $28^oW$) so the distance to the WA shelf
break is enough to fit a RW wavelength in the sector where PV gradients are mainly directed from south to north and relatively
uniform spatially. Nevertheless, we suspect that the differences in PV gradients between layers 1-2 and the deeper layer are
implicated in the RW signal attenuation observed within 1000 km from shore (Fig. 8 and 16). One alternative explanation
would be that linear waves satisfying (6) are evanescent, but this is inconsistent with the critical latitude found for baroclinic
mode 1 semi-annual waves $(22^oN)$[5]. Most importantly, the width of the sector situated east of the zero PV gradient decreases
rapidly with latitude and is only $\sim 300$ km off the coast at $18^oN$, which precludes the existence of the weakly dispersive
semi-annual RWs found further south.

Overall, the modifications of the ETNA PV field by vortex stretching effects in the density range 25.2-26.7, where most of the
meridional flow and Rossby wave energy are concentrated, appears as a good candidate to explain the (meridionally variable)
cross-shore damping scale for RWs and the progressive reduction of RW amplitude north of $15^oN$. At deeper depth, below
300-400 m, across-shore sections reveals alternating bands of poleward and meridional flow that tend to migrate westward
thereby suggesting the presence of Rossby wave activity (not shown), in agreement with the observational findings of Hagen
(2001). The physical processes responsible for the particular PV structure present in the upper ETNA will be discussed in Part
2.

Finally, note that semi-annual Rossby wave dynamics may not solely be an offshore consequence of WABC variability. It
can indeed result from direct wind stress curl forcing, and, in turn, contribute to forcing the boundary current seasonal cycle.
Although semi-annual variability of the wind curl signal is weak in this sector (Fig. 11) some resonant excitation of free Rossby
wave mode having a nearshore signature presumably contributes to the WABC semi-annual variability. Between 10 and $20^oN$,

---

[5]Conversely, the Rossby wave signal described by Gómez-Valdivia et al. (2017) in the North-East Pacific off Baja California at latitude comprised between
25 and $30^oN$ must be evanescent. We note that the across-shore scale over which it decays (Fig. 7b in Gómez-Valdivia et al., 2017) does not seem very different
from the one we find at $14^oN$.





the wind stress curl/Sverdrup transport fields shown in Fig. 1) exhibit across-shore spatial variations with a contribution of wavelengths $\approx$ 500-1000 km. This is in the range suited to excite baroclinic mode 1 Rossby waves. Incidentally, no coastal source of energy is apparent in the horizontal distribution of semi-annual baroclinic mode 1 kinetic energy (Fig. 16a).

## 7 Discussion and conclusions

An eddy permitting numerical simulation with realistic forcings has been analyzed to investigate the dynamics of the boundary current along the West African seaboard. The depth range of interest was chosen to be above the $\sigma_t = 26.7$, which broadly coincides with the upper 250 m of the water column and places the focus on the layer of fluid where the wind-driven circulation is overwhelmingly concentrated. The precise geographical focus is on the southern sector of the Canary current system between $\sim$ 10 and 20$^o$N. In this area wind stress curl (both nearshore and offshore) is robustly positive, *i.e.,* conducive to poleward

flow[6]. In fact, upper ocean equatorward currents are rarely found over the WA continental slope. The model poleward flow is characterized by two main intensification periods in spring and fall, that we interpret as the consequence of low-frequency coastal trapped wave activity generated by seasonal wind fluctuations along the African shores, locally and remotely[7]

Despite some differences in their forcing regions and precise depth/latitude range of influence, the spring and fall model WABC intensifications bear important similarities. First, they are the two peaks of a semi-annual cycle that nearshore WSC

contributes to reinforce through excitation of the associated free Rossby wave mode. This process must be of modest importance though, given the boundary intensification of the ETNA currents seen throughout the study. Considered in isolation, the spring intensification accompanies the relaxation of winter-time coastal upwelling winds in the latitude range 7-20$^o$N, as the ITCZ shifts northward toward that area. Flow intensification is found in the subsurface and is broadly consistent with the theoretical framework relevant to the dynamics of undercurrents in upwelling systems. The fall intensification has a more remote origin that

we are able to locate in the Guinea Gulf, through spatio-temporal analyses of both wind forcing and coastal ocean response. In that respect, this work tends to substantiate old assertions about the connection between the boundary current flowing offshore of Senegal/Mauritania and poleward flow in the Guinea Gulf, albeit only during part of the annual cycle. Conversely, and in contrast to what has been hypothesized for the southeast tropical Atlantic (Schouten et al., 2005; Rouault, 2012), wind variability and Kelvin wave activity in the equatorial Atlantic are not found to be implicated in the forcing of the WABC

semi-annual cycle (McCreary et al., 1984). Our results also differ from those for the eastern South Atlantic (Junker et al., 2015) in that the seasonal cycle of the WABC is not directly linked with the local wind stress curl, which has no semi-annual modulation.

---

[6]This is in contrast to most other eastern boundary upwelling system where offshore and nearshore wind stress curl tend to be of opposite sign (Bakun and Nelson, 1991)

[7]In contradiction to the assertions made in various places including Mittelstaedt (1991) we were unable to establish a connection between the summer-time model NECC pulse and the fall near-surface intensification of the WABC. Our unsuccessful attempts included diagnostics aimed at tracking the propagation/advection of patterns of elevated surface pressure signals from the region 23$^o$,8-10$^o$N (where the northern NECC summer pulse is strongly felt; Fig. 6) toward the east-northeast where they could contribute to enhancing alongshore pressure gradients at the WA continental slope. In Part2 we will show that time scales associated with advection are too long for this to happen (Rossby waves propagate pressure signals toward the west and are therefore no candidate).





More quantitatively, the model provides estimates for the poleward transport over the WA continental slope. They depend somewhat on the precise choices made for the control surface (depth/across-shore integration bounds and position in latitude). At the latitude of Senegal (resp. Mauritania), geostrophic transport above $\sigma_t = 26.7$ is of the order of 1 Sv (resp. 0.4 Sv), *i.e.,* a large (moderate) fraction of the theoretical barotropic Sverdrup transport. We relate this to the meridional changes in WABC

dispersion through Rossby wave generation. Indeed, dispersion is most pronounced at low latitudes where Rossby waves travel faster and higher baroclinic modes can be impacted. The vertical structure of the boundary currents reflect these differences. Upper ocean confinement of the Sverdrup flow by Rossby waves (Anderson and Gill, 1975; Philander and Yoon, 1982) is systematically most pronounced at lower latitudes, hence the north-south differences in depth range of the WA boundary current.

The model analysis of Rossby wave activity reveals important differences with previous descriptions of these waves, carried out at larger scale in the eastern/central North Atlantic (Hagen, 2005). Most notably, we find that Rossby waves generated at the eastern boundary remain confined into a well-defined ocean sector delimited by the WA seaboard and an oblique line running north-east/south-west where background potential vorticity gradients vanish in the upper ocean. Such modification of the potential vorticity field is due to stratification effects having a magnitude comparable to planetary vorticity effects ($\beta$).

Past studies of Rossby wave activity in the North Atlantic have classically been made using linear 1 1/2 layer reduced gravity models (Da Silva and Chang, 2004; Garzoli and Katz, 1983; Busalacchi and Picaut, 1983) with no background flow that RWs can interact with. It turns out to be an important limitation for the ETNA sector we considered here.

The relevance of this numerical investigation to the real WA ocean is an obvious concern. Although TROP025 skills at regional and basin scale have been demonstrated (Sec. 2) model biases cannot be excluded at the scale of the WABC. Because

there have been relatively few observational programs in this part of the world ocean we can only offer limited and qualitative insight into model realism, for example on the reality of the two WABC intensification phases and the associated flow characteristics. Although the existence of two poleward intensification phases is not systematically recognized in previous studies published observations are not inconsistent with the model behavior, including on the timing of these two phases.

The recent field experiment CANOA08 took place in November 2008 at a time of year where the poleward flow should

be most intense including near the surface. Above $\sigma_t = 26.85$ the flow over the continental slope carried SACW up to 25$^o$N (Peña-Izquierdo et al., 2012) where vanishing meridional transports were found, *i.e.,* exactly the November climatological limit determined for the same density class in the model (not shown). In addition, observed transports are broadly consistent with those found in TROP025 particularly for the uppermost stratum examined by Peña-Izquierdo et al. (2012) (see model-data comparison in table 1). In CANOA08 the intense poleward surface and subsurface currents found in the vicinity of

Cape Blanc at surface and subsurface levels are interpreted by the authors as, respectively, "a late expression of the summer Mauritania current" and a local response to strong upwelling winds (Peña-Izquierdo et al., 2012). Our model results cast doubt on these interpretations and suggest, instead, that CANOA08 may have sampled the ETNA at the (normal) time when the fall intensification of the surface and subsurface WABC is remotely forced.

On the other hand, comparison with CANOA08 observations raise some concerns about the alongshore continuity/coherency

of water mass transport in TROP025. In the model, the low salinity signal characteristic of SACW does not penetrate north of





Cape Blanc irrespective of the season (Fig. 9). This is inconsistent with observations reported in Peña-Izquierdo et al. (2012) (as well as older ones) showing large amounts of low salinity SACW up to 24$^o$N. The across-shore exchanges of water between the WABC core and the open ocean may thus be overestimated in the model, plausibly as a consequence of insufficient horizontal resolution. Given the difficulty to maintain moorings in this area multi-year repeats of the CANOA array at different seasons

would provide useful information on the temporal variability of the WABC.

To our knowledge, the only data that are available to assess model realism during the spring WABC intensification are those carried out at 21$^o$40'N as part of the JOINT-1 experiment to investigate shelf and slope currents in the vicinity of Cape Blanc. Currentmeter data (Mittelstaedt et al., 1975, their figures 4 and 6) underscore the importance of synoptic variability with dramatic fluctuations of the poleward flow on time scales of days to weeks. Consequently, JOINT-1 efforts remain inconclusive

with respect to the mean structure of the boundary currents at that time of year (including on the possible existence of a near-surface poleward countercurrent over the continental slope). A similar issue plagues the study of the drift of a parachute drogue released off Cape Blanc at 50 m depth (Hughes and Barton, 1974). The trajectory reveals intense northward flow (24 cm s$^{-1}$ over a 12 hour period) that cannot be considered representative of average conditions. However, combined with the CINECA Feb.-March 1972 velocity observations at 19-20$^o$N (Mittelstaedt, 1976, his figure 9) showing a core of poleward velocities in

excess of 8-10 cm s$^{-1}$ in the depth range 100-200 m, the general impression is that TROP025 underestimates the intensity of the flow at these latitudes (based on Fig. 9 and also on the examination of a figure similar to Fig. 7 for the latitude band 19-21$^o$N, not shown). This would be another possible reason why the low salinity signal associated with the presence of SACW north of Cape Blanc are not reproduced in TROP025.

Overall, the realism of the model boundary circulation is uncertain given the scarcity of available observations. In addition,

our dynamical interpretations frequently invoke baroclinic mode decomposition which are not strictly valid in the horizontally heterogeneous conditions where we use it. More elaborate approaches such as WKB ray tracing may prove useful in this regard, *e.g.,* to clarify the reasons why sea level and upper ocean flow signals propagate offshore at different speeds. In this context, the present study should be seen as a way to stimulate and guide future work in this highly undersampled part of the world ocean. Part 2 which aims to connect the WABC to the regional circulation context of the subtropical North Atlantic shadow

zone shares the same general objective.

*Acknowledgements.* While conducting this research L. K. was funded through the ART PhD program of the Institut de Recherche pour le Développement. We thank A. Colin de Verdière, B. LeCann, R. Schopp, J. Sirven, and J. Gourrion for useful discussions and comments.



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




**Figure 1.** *Seasonal climatology of Sverdrup transport ($[m^2\ s^{-1}]$, excluding the Ekman flow in the surface layer) computed from the DFS5.2 wind forcing fields averaged for Jan.-March (upper left), April-June (upper right), July-Sept. (lower left) and Oct.-Dec. (lower right). The main regional flow/thermohaline features and capes are indicated in the upper left panel: the Canary current (CC), North Equatorial Current (NEC), North Equatorial Counter-Current (NECC), North Equatorial Under-Current (NEUC), Guinea Dome (GD), Cape Verde Frontal Zone (CVFZ) which separates the realm of the North Atlantic Central Waters (NACW) and South Atlantic Central Waters (SACW). The TROP025 100 m isobath along which several analyses are made is shown in black. Three geographical limits used to define the integrated upwelling indices computed in Sec. 5 are shown with blue dots.*

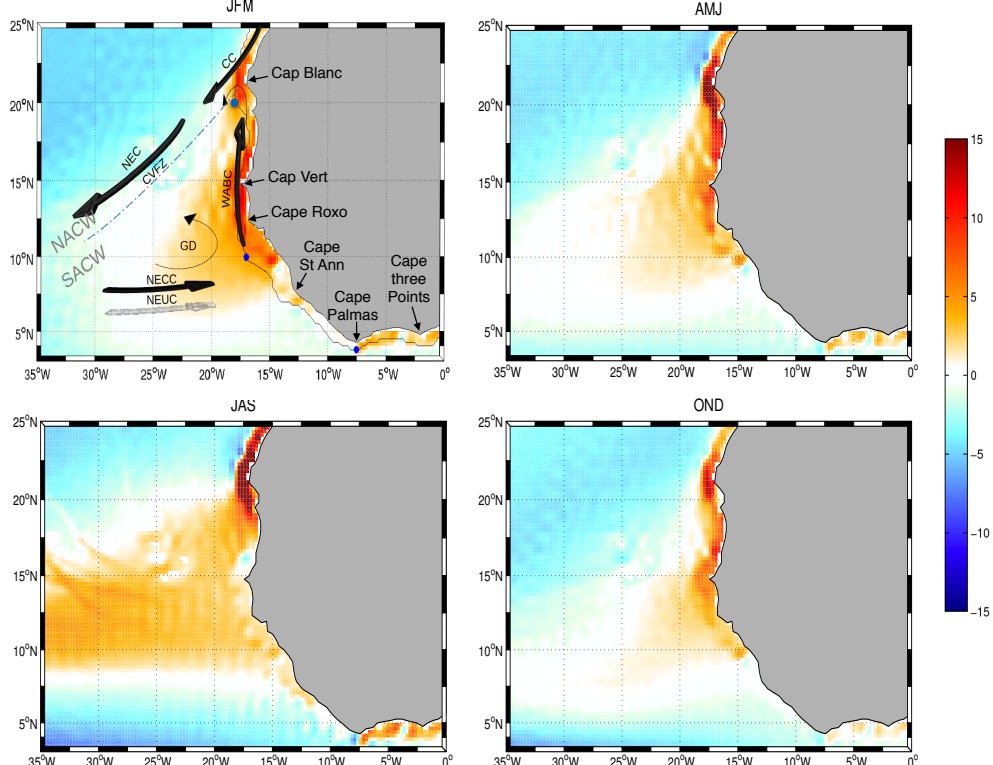



**Figure 2.** *Meridional (resp. zonal) section at 26°W (top) (resp. 13°N; bottom) for temperature [°C] averaged during september-october. Left (resp. right) panels are for observations (resp. model).*

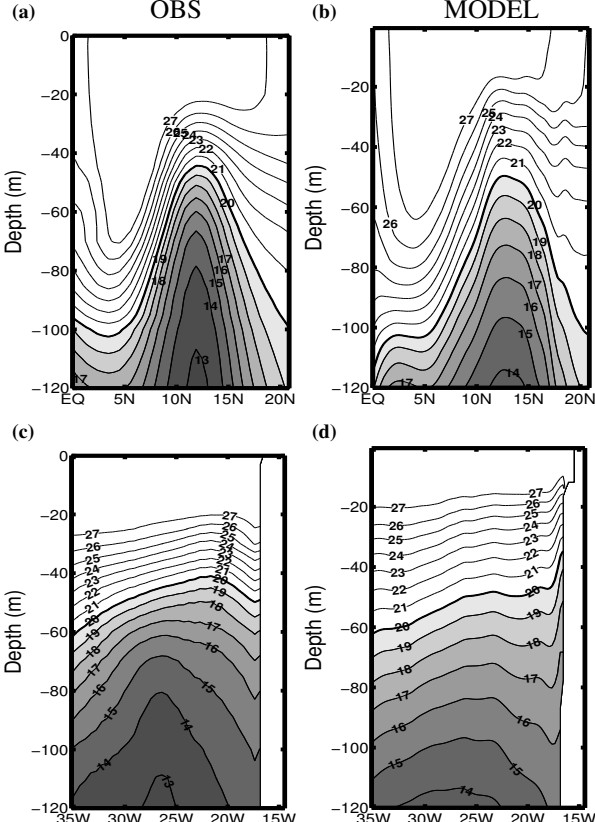




**Figure 3.** *Summer (top; June.-Sept.) and winter (bottom; Dec.-Feb.) mean dynamic height at 50 m relative to 500 m [$m^2\ s^{-2}$]. The contour interval is 1 $m^2\ s^{-2}$. The associated geostrophic circulation is also shown (vectors). Left (resp. right) panels are for observations (resp. TROP025).*

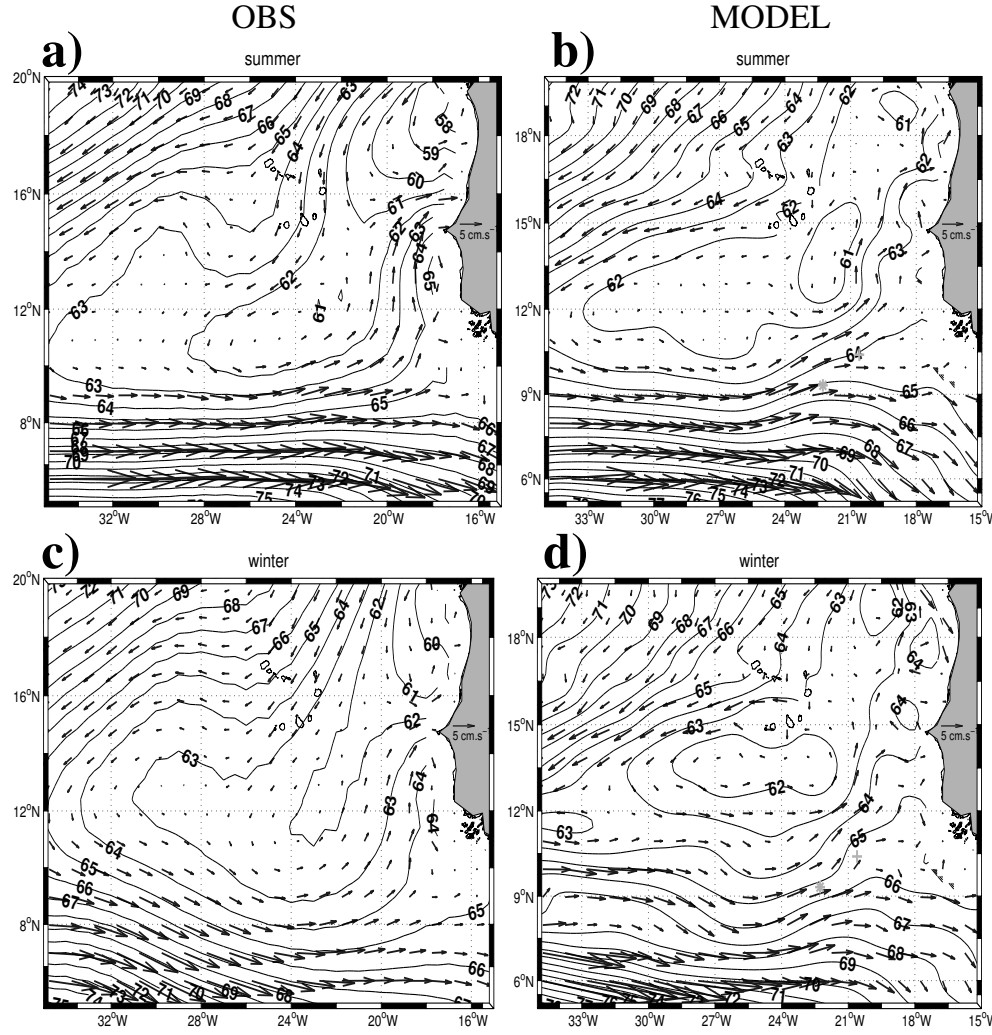



**Figure 4.** *Depth [m] of the isopycnal density surface $\sigma_t = 25.2$ (top) and $\sigma_t = 26.7$ (bottom) in CORA (left) and TROP025 (right).*

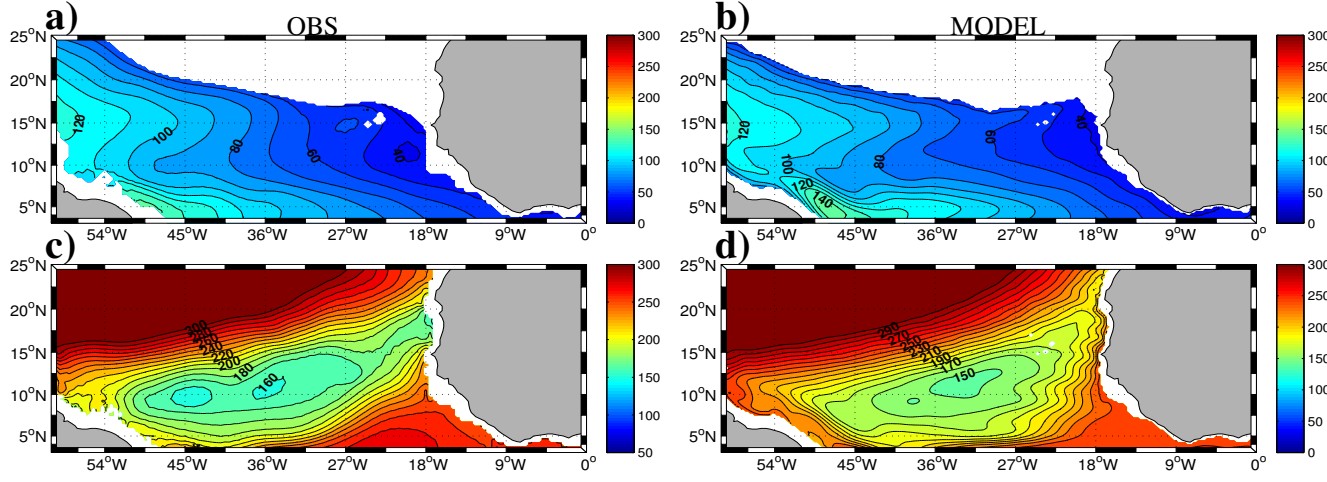

**Figure 5.** *Time-depth representation of TROP025 climatological temperature [$^oC$] at $4^oN$-$5^oW$. This figure is to be compared with Fig. 15b in Picaut (1983)*

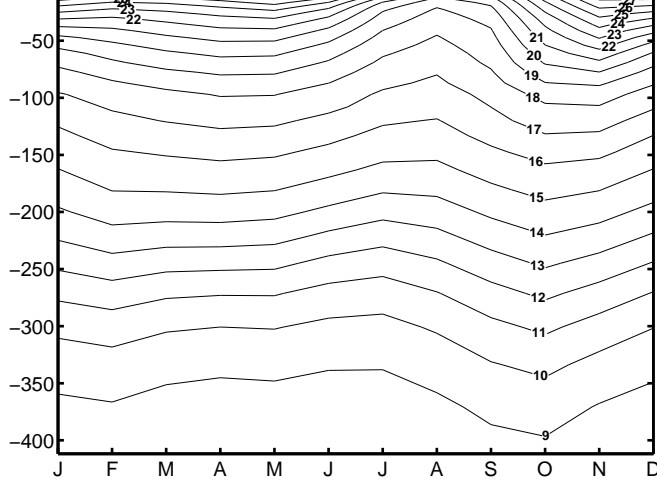



**Figure 6.** *Model monthly climatology of zonal velocity at the ocean surface [m s$^{-1}$]. This figure shall be compared with Fig. 6 in Rosell-Fieschi et al. (2015)*

**Figure 7.** *Seasonally averaged vertical-zonal section of meridional velocity (in colors; [cm s⁻¹]), $\sigma_t$ (blue lines; [kg m⁻³]) and mixed layer depth (black line; [m]) averaged between 13° and 15° N.*



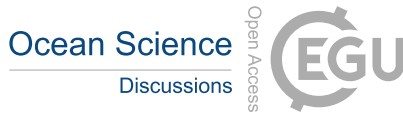

**Figure 8.** *Left: time-longitude diagram of vertically integrated meridional transport [$m^2\ s^{-1}$]. Propagation at the theoretical phase speed of non dispersive semi annual Rossby waves (4.6 cm $s^{-1}$) and at 3.5 cm $s^{-1}$ are respectively shown with dotted and dashed lines. The latter value is the one we choose as most consistent to describe the propagation of spatio-temporal patterns in the diagram. Right: climatology of meridional transport integrated vertically and across-shore. Vertical integration is performed from the isopycnal surface $\sigma_\theta$ = 26.7 up to the bottom of the mixed layer (left panel and red curve in right panel) or up to the surface but excluding Ekman transport, i.e., we only take into account the geostrophic flow (blue curve in right panel). Across-shore integration is performed from the shoreline to the first location where vertically integrated transport vanishes.*

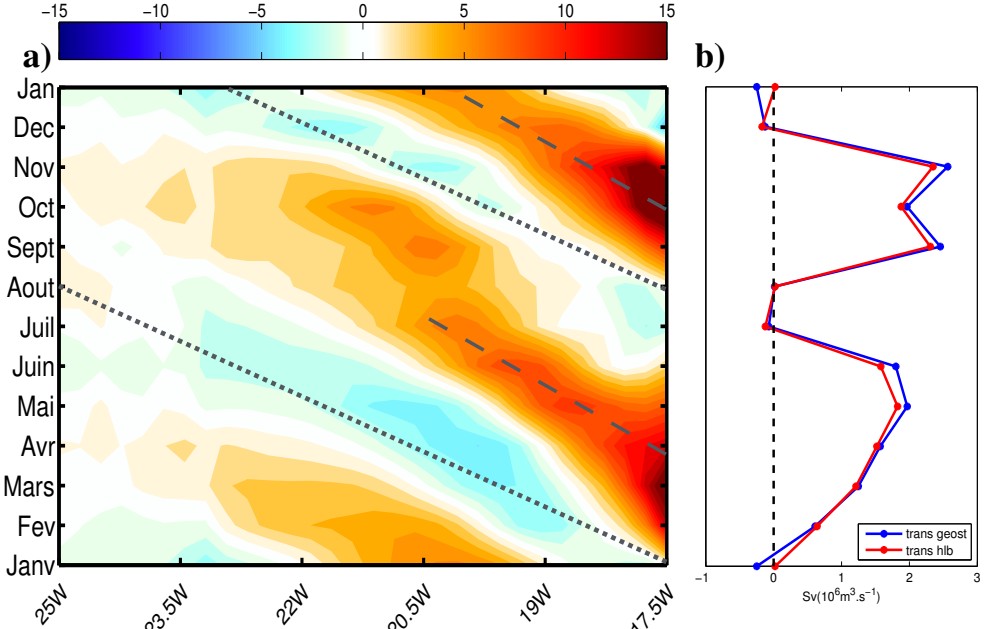

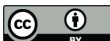


**Figure 9.** *Seasonally averaged vertical section of along-shore velocity (in colors; [cm s$^{-1}$]), $\sigma_t$ (blue lines; [kg m$^{-3}$]), and mixed layer depth (black line; [m]) following the shelf break (across-shore averaging between the 100m isobath and 6 grid points - 150 km - offshore.*




**Figure 10.** *Monthly climatology of meridional transport ($[m^2\ s^{-1}]$) integrated between the isopycnal surfaces $\sigma_t = 26.7$ and the surface, excluding wind driven Ekman transport calculated from TROP025 wind fields. The two thin black lines represent the location where zero PV gradients are found in Fig. 17.*

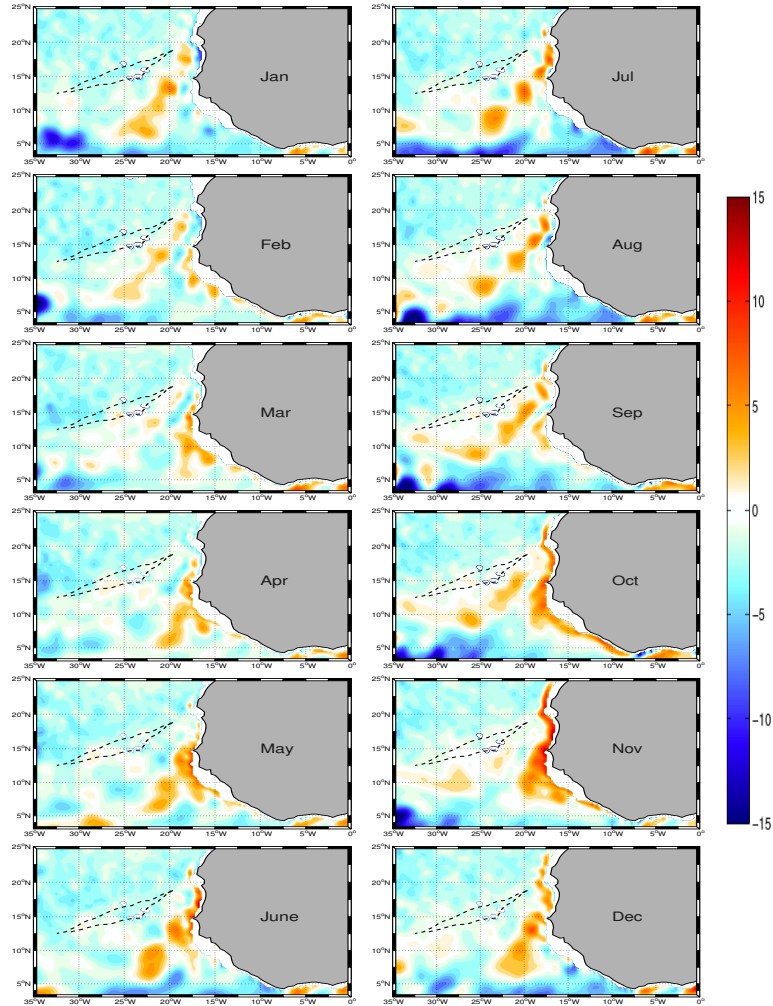





**Figure 11.** *Seasonal cycle of Sverdrup transport $V_{sv} = \frac{f}{\beta} curl(\frac{\tau}{\rho_0 f})$ (red lines) and meridional geostrophic transport vertically integrated between $\sigma_t$ = 26.7 and the surface (blue lines). Both transports are across-shore averaged between the 100 m isobath and 6 grid points offshore. Solid lines are for the Cap Vert region ($13^oN$-$15^oN$) while dashed lines are for the Cap Blanc region ($19^oN$-$21^oN$)*

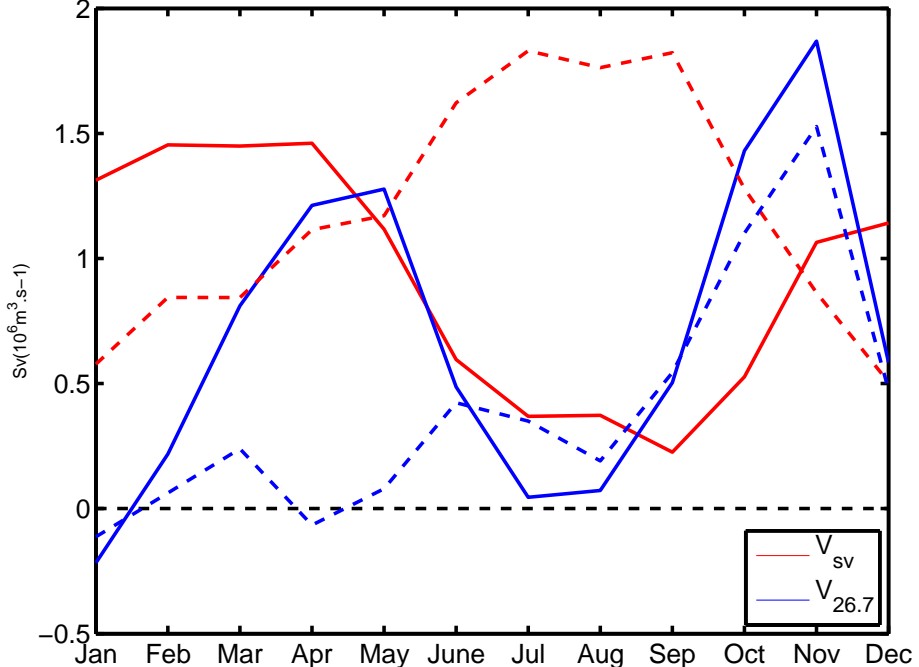





**Figure 12.** *Time-alongshore distance diagram following the 100 m isobath (see location in Fig. 1 for the TROP025 climatological seasonal of a) depth anomaly of the $18^o$ isotherm $z_{18}$ (colors, [m]) and SLA (contours, [cm]), (b) alongshore geostrophic transport between $\sigma_t = 26.7$ and the surface $V_g^{26.7}$ [$m^2\ s^{-1}$] c) across-shore Ekman transport anomaly Oblique solid black lines correspond to propagation speeds of 0.2 (Feb.-June) and 0.9 $ms^{-1}$ (Aug.-Jan.). In panel c, vertical gray lines delineate the sectors over which sector upwelling indices are computed (see Fig. 13). Power spectral densities associated with the annual and semi-annual harmonics of the Ekman transport along the 100 m isobath are displayed above panel c. Oblique lines are subjectively drawn in panel a (and repeated in panels b and c) to indicate where spatio-temporal patterns are suggestive of propagation.*

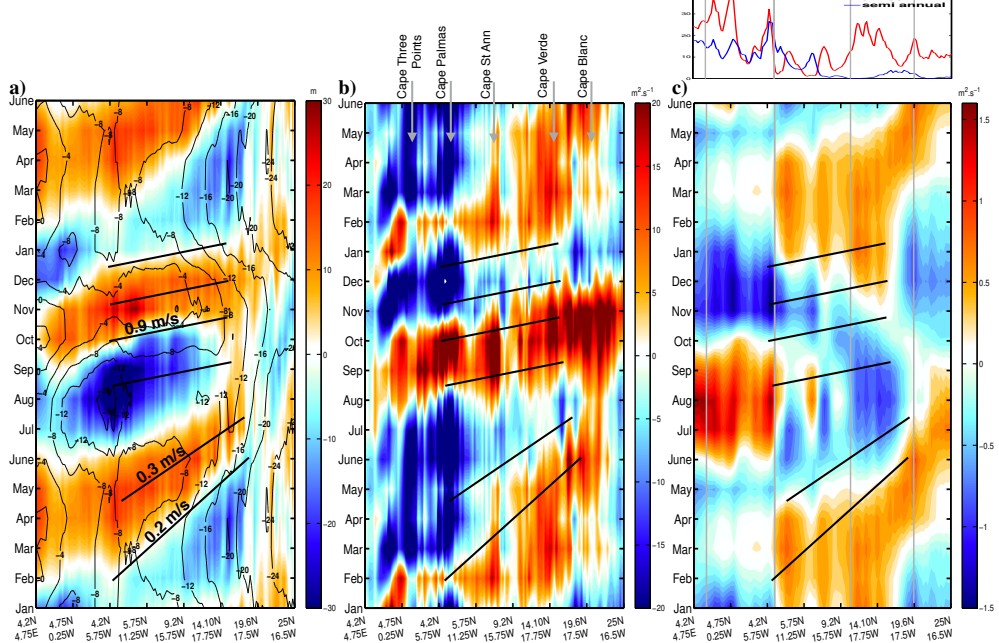



**Figure 13.** *Monthly mean climatology of across-shore Ekman transport [Sv] integrated along the 100 m isobath for three sectors: in the Guinea Gulf between longitudes 3.5°E and 7.5°W (black), further west between 7.5°W and 17°W (blue); and between 10 and 20°N (red). The position of these sectors which are of comparable length is indicated in Fig. 1a and 12c.*

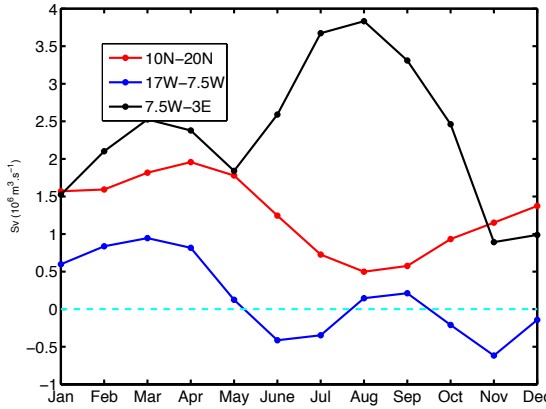

**Figure 14.** *Wavenumber-frequency dispersion diagram for baroclinic Rossby waves corresponding to values of the deformation radius $R_d$ = 20, 30, 40, 50 and 60 km (solid lines with $R_d$ values increasing upward). The dispersion curve for $R_d$= 30 km (resp. $R_d$=50 km) is in black (resp. thick black). The black (resp. gray) filled dot and dashed line represent the theoretical (resp. model) Rossby wave characteristics: $\omega$=2 $\pi$/(6 months); k = - 2 $\pi$/(731 km); c = 4.6 cm s$^{-1}$ (resp. $\omega$=2 $\pi$/(6 months); k = - 2 $\pi$/(650 km); c = 3.5 cm s$^{-1}$, estimated in Fig. 8). The horizontal dotted line corresponds to the semi-annual frequency.*

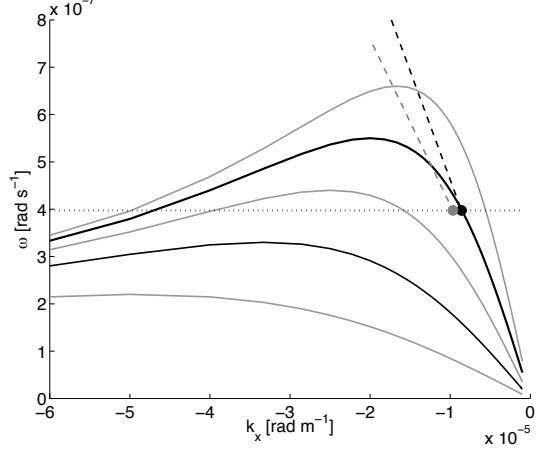





**Figure 15.** *Vertical structure of the ETNA baroclinic modes 1 to 3 for pressure and horizontal velocities (upper 2500 m only). Calculation is made using TROP025 stratification at $14^oN$, $18^o30'W$. The associated reduced gravity phase speed ce [m s$^{-1}$] for each mode is also indicated.*

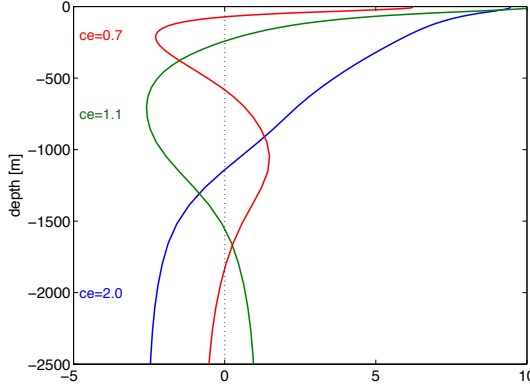

**Figure 16.** *Vertically integrated kinetic energy [m$^3$ s$^{-2}$] for the semi-annual meridional velocity decomposed onto baroclinic modes. Only the first four modes are shown. Time averaging is performed over the semi-annual cycle. The two white dashed lines represent the location where zero PV gradients are found in Fig. 17*

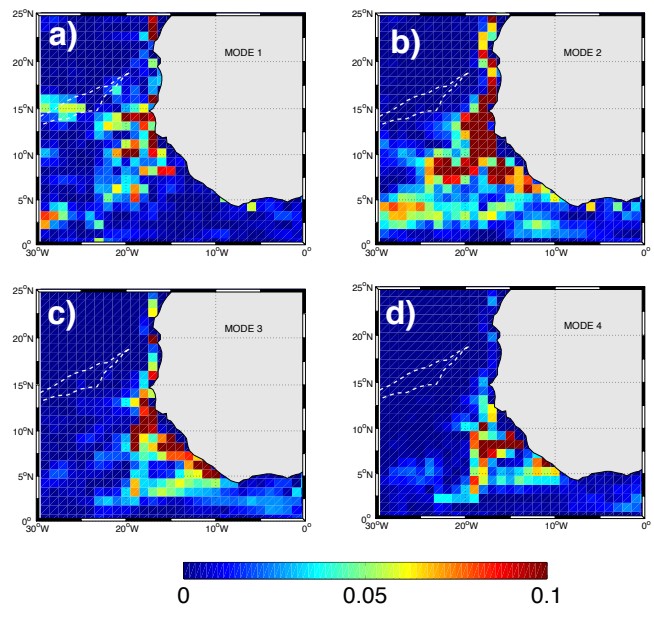




**Figure 17.** *Shallow-water potential vorticity field in the ETNA computed from TROP025 (color; $[10^{-8}\ m^{-1}\ s^{-1}]$) using (4) for the density classes 25.2-26.3 (top), 26.3-26.7 and 26.7-26.9. These values are such that the three layers are of comparable thickness and the same colorscale can be used. Potential vorticity gradients are also shown in vectors. Note the vanishing gradients found along the white dashed lines in top and middle panels (the position of the two lines differ slightly, e.g., see their position with respect to the Cape Verde islands).*

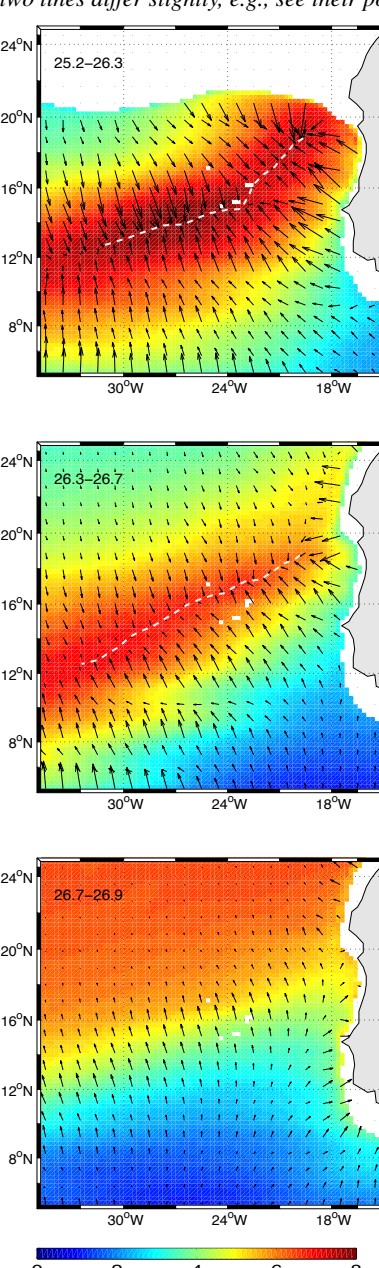





**Table 1.** Climatological Sverdrup transport $T_{sv}$ (geostrophic part, see Sec. 2) computed from DFS5.2 winds at two different latitudes over the across-shore band from the shelf break to 150 km offshore. Model geostrophic transports computed over the same across-shore areas are indicated as a percentage of $T_{sv}$, for three different ranges of vertical integration from the surface to $\sigma_t = 26.7$ ($V_g^{26.7}$, third column), 500 m ($V^{500m}$, fourth column) or 1000 m ($V^{1000m}$, fifth column).

|  | $T_{sv}$ (Sv) | $V_g^{26.7}$ (%) | $V^{500m}$(%) | $V^{1000m}$ (%) |
|---|---|---|---|---|
| 14°N | 0.92 | 75 | 80 | 79 |
| 20°N | 1.2 | 34 | 59 | 69 |

**Table 2.** Meridional transports observed in Nov. 2008 reported in Peña-Izquierdo et al. (2012) (left) and their TROP025 climatological equivalent for the month of November (right) separated by a "/". Transport values are provided for two different density layers (SW for surface waters, $\sigma_t < 26.46$; uCW for upper central waters, $26.46 < \sigma_t < 26.85$) and four latitudes (16.25, 17.5, 20 and 24°W). In situ values are estimated visually from Fig. 7a-c in Peña-Izquierdo et al. (2012) for the cross-sectional area between the shelf break to 60 km offshore. For the model, transport is computed over four grid cells (100 km) situated offshore of the shelf break. Using this wider across-shore section is meant to account for our limited horizontal resolution but transports estimated over only three grid cells only differ by 15-20 %.

| SW - 16.25 | SW - 17.5 | SW - 20 | SW - 24 |
|---|---|---|---|
| 1.3 / 1.0 | 1.8 / 1.4 | 1.0 / 1.2 | 1.0 / 0.6 |

| uCW - 16.25 | uCW - 17.5 | uCW - 20 | uCW - 24 |
|---|---|---|---|
| 1.0 / 0.3 | 0.6 / 0.6 | 0.6 / 0.7 | 0.7 / 0.4 |