# Peer review of "A model perspective on the dynamics of the shadow zone of the eastern tropical North Atlantic. Part 1: the poleward slope currents along West Africa"

_Ocean Science, 2018_

## Referee Comment (RC1) · Anonymous Referee #1 · 18 Apr 2018

This interesting paper is a welcome attempt to cast light into the "shadow zone" of the region equatorward of the Canary Current's separation from the African coast by examining the regional dynamics with the aid of a numerical eddy-resolving model. This is an area with relatively sparse observations even during the period in the 1970s when the adjacent northwest African upwelling was subject to considerable international interest. The focus is on the spatial and seasonal variability of the poleward flow that has been reported throughout the region but never subject to systematic observation over time. Both local and remote wind forcing are found to be instrumental in driving the flow, but vary in relative importance between spring and autumn. The model indicates a westward propagation of the poleward flow, at least south of Cap Blanc, consistent

with Rossby wave dynamics. The thorough and coherent approach of the analysis progresses our understanding of the region, although many uncertainties remain. The authors provide a caveat in their concluding section that the model fails to reproduce the well-established penetration of the SACW mass to the north of Cap Blanc. As they also point out in the validation section, the model does not convincingly reproduce the zonal structure of the Guinea Dome as indicated by hydrographic data. Although both inconsistencies may reflect inadequate horizontal resolution, or other problem, of the model, it is also true that observations of the hydrography and current field are far from sufficient to provide realistic comparisons with the model. So clearly, much work remains.

While in general the writing is clear and correct, a rather relaxed and somewhat conversational style leads to lengthy text that could be made more terse, with a modicum of effort. Examples include "are quite noisy but the overall impression is that they tend to be more toward the west... p7 l24) or "note also that significant small-scale noise results from the sinuosity of the flow past topographic irregularities whose position can thus locally depart from the shelf break, e.g. in relation to standing meanders in the lee of headlands, as noticeable in ... p13 l10". Quite so, but it could surely be said more succinctly.

Figures are inconsistently labelled, some with panels marked by letter, others not. Even when so labelled, the panels are in some cases referred to not by letter but by position.

Overall the paper is significant, of excellent quality, and well prepared and the scientific development is sound. If the authors could tighten up the text a little, and attend to the few specific points mentioned above and the details below, the paper is recommended for publication as an innovative and useful contribution.

Abstract: p1 l4 "a suite of" is unnecessary l6 "southernmost" rather than "outmost"

Introduction: p2 l5 preferable to refer to "northwest Africa" for generality and to avoid controversy with respect to the internationally disputed borders in the area. p2 l5

Delete"see geographical and oceanographical" p2 l9-10 The water mass front and its weakening with depth have been recognized long before the work cited, e.g. Allain (1970), Fraga (1973) and other papers of that period, and more over later years. P4 l19 "literature" not "litterature"

Model evaluation: p7 l16 Figure 6 is called out of order, before Figure 3. p8 ll5-12 The doming structure on the zonal line in Fig 2 is weaker in the model. The localized uplift at almost all levels near coast is not mentioned. Incidentally although the figure panels are labelled a-d, the legend refers to left,right, top and bottom.

Seasonal cycle of the WABC: p10 l10 State "Alongslope vertical sections of ..." to emphasize the difference with Figure 7. p10 l11 "across-shore averaging" would be better expressed as "across-slope averaging" as the shore bounds the flow. This occurs in a few other places in the text.

WABC coastal dynamics: p11 l10 "as more classically estimated" - does this mean calculated from the observations? If so, please state it more directly. P12 l15 Delete "it" before "as occurs".

---

## Referee Comment (RC2) · Anonymous Referee #2 · 1 May 2018

This paper investigates the forcing mechanism of the poleward undercurrent at annual and semi-annual frequencies, focusing on the region between 10°N and 20°N. Four distinct processes are looked at: i) local generation of a poleward undercurrent in conjunction with coastal upwelling conditions ii) remote forcing of poleward flow with subsequent propagation in the form of coastal trapped waves iii) local modulation of the nearshore Sverdrup transport in relation with the seasonal cycle of the wind stress curl iv) Rossby wave modes at the semi-annual frequency. The analysis is based on a forced experiment with an OGCM at the resolution $\frac{1}{4}°$ over the period 1979-2015.

Considering the scarcity of observations in that region, the model simulation is viewed

as material for improving our knowledge of the boundary current dynamics in this region at seasonal frequency, recognizing that the model set up has certainly limitations (e.g. resolution) to address all aspects of the variability. The analysis provides interesting insights in the variability in this region. It conveys in particular the idea that the westward propagation of energy away from the coastal guide is an important process contributing to the poleward attenuation of the boundary current. This westward propagation of energy is interpreted as resulting from the propagation of a semi-annual/annual extra-tropical Rossby wave, although the agreement between model and linear theory indicates that non-linear dynamics is certainly involved. The analysis also suggests that the poleward flow in the southern part of the Canary current system has a remote component originating from the Gulf of Guinea (i.e. wind forced), which contrasts with former studies that invoke an equatorial origin.

The analysis is based on the estimate of Sverdrup transport and geostrophic meridional transport within a priori determined depth ranges, so as to discriminate locally wind forced process and remote effects (equatorial origin or along the coastal wave guide). It makes use of the linear theory for the interpretation of the variability.

The paper is well written and provides a nice overview of all the potentially important processes in this region that are tested based on the medium resolution model simulation, offering a benchmark for other model analysis and material for the interpretation of data. While it is useful to have this broad perspective of the variability, the caveat is that it feels sometimes that the paper lacks focus making it difficult to retain the main result. I think this problem can be easily overcome through improvement of the presentation. For instance, the main conclusions could be clarified through providing a schematic summarizing the main processes and highlighting the findings of the paper. It might also be useful to summarize in a table the main processes that have been tested, through which diagnostics, and the consistency with previous studies.

They are also in some instances in the text some unclear (or too vague) statements (see specific comments).

General comments:

1) The title should reflect that the paper mostly investigates the seasonal variability of the poleward flow. In fact the paper appears to me as a study of the forcing mechanism of the semi-annual variability more than a study of the dynamics of the shadow zone which refers to the mean deep circulation?

2) The section 6 is devoted to the analysis of the meridional flow seasonal variability in terms of extra-tropical Rossby wave. On the one hand the authors suggest that a semi-annual Rossby wave can radiate off-shore at latitudes South of $\sim 15°$N but in Section 5 they also show that there is a possible remote source of semi-annual poleward flow off WA, implying the propagation of Coastal Trapped Waves at the semi-annual frequency (forced by the winds along the coast of Ivory coast). In the frame of linear theory, at a given frequency, a wave is either trapped along the coast or radiates off-shore, so could you try to reconcile these apparent conflicting results (or clarify the text). This calls also to clarifying what is the forcing mechanism of the semi-annual Rossby wave that is discussed.

Other comments:

p. 2, l. 28-29: "in part because the shape of the African continent produces a curvature of the trade winds" through which process?

p.3, l. 4: AWA not defined

p. 9, l.4-5: Not clear here if the amplitude the semi-annual and annual cycles was actually estimated in observations and model. How do you estimate the 50%?. It would be useful to indicate an error bar (dispersion) associated to the number of years that is retained to calculate the mean of Fig. 5 considering that the observations of Picaut corresponds to the period 57-64. You could select randomly chunks of 8 years in the model and provide the dispersion among the ensemble chunks. This would inform also on the possible influence of decadal variability on the validation.

p. 9, l. 7-9: "To put these biases into perspective it would be useful to know the degree to which the observations in Picaut (1983) are representative of a long alongshore stretch of ocean, as opposed to very local conditions that TROP025 cannot represent due to lack of resolution." Not clear. Do you mean that the observations of Picaut (1983) would account for coastal trapped wave variability, which may not be well simulated in TROP025 owing to the too coarse resolution? What is the critical latitude of the annual and semi-annual period? It could be useful to mention this information at that stage for clarity.

Figure 8: indicate latitude ranges? We assume it is the same than Figure 7?

p. 9, l. 28-30: "In winter and summer a core of poleward flow present a few hundred kilometers from shore is suggestive of westward propagation of the poleward undercurrents." This is not straightforward. It is more like there is an annual Rossby wave that is in phase with the seasonal cycle of the undercurrent? The concept of an undercurrent propagating off-shore as a Rossby wave is not straightforward since the undercurrent is usually tight to the bottom (slope) boundary layer dynamics. The text may need to be clarified.

p. 10, l. 5-6 "Zonal integration is performed from the coast to the first offshore location where the flow changes 5 direction, so the width over which this transport is achieved varies in time." Is it possible to plot the distance from the coast over which the zonal integration is performed on Figure 8a.

p. 10, l. 27-31: You could do maps of the annual (semi-annual) amplitude and phase of the meridional transport of Figure 10 in order to support the interpretation and provide more quantitative statement.

p.11, l. 30.31: It is not clear why it should be "resonant excitation of free Rossby wave modes". It could be locally forced Rossby waves or remotely forced Rossby waves? Can we have purely "resonant" extra-tropical Rossby waves at these latitudes without coupling with wind stress? Any reference to support such a hypothesis considering

that the references that are provided are for resonant modes in the equatorial wave guide?

p.14, l. 20-21: "With respect to 4), z18 and V26.7,g are not precisely in phase as they are expected to be for theoretical Kelvin waves (Cushman-Roisin and Beckers, 2011)". This assumes a 1.5-layer (i.e. one barocline mode) dynamics. In a multimode context you can have a phase difference.

p. 14, l. 24 :" Due to Rossby waves pressure fluctuations associated with CTWs propagate offshore" This is not clear. Please rephrase.

p.15, l. 15: "...the presence of a semi-annual Rossby wave coupled to the coastal trapped wave activity". This is unclear. Do you mean the coastal trapped wave activity is concomitant with the semi-annual Rossby wave. From a theoretical point of view, you cannot have a Rossby wave and a coastal trapped wave at the same frequency. The wave is either trapped or radiate as Rossby wave.

p.15, l. 20: "Rn= 50 km which is approximately the value we find at $14°N,18,30°W$ for the first deformation radius, based on the model stratification". How do you calculate Cn in the model? More details on the method used to derive the baroclinic mode structure should be provided. In particular, do you use "continuous" (interpolated) profiles or does the modes are derived on the model vertical grid?

p. 16, l. 1.: "integration to a layer in which the poleward flow is concentrated, e.g., the upper 200 m, leads to similar results". This is surprising since the baroclinic modes are no longer orthogonal when integrated over such a shallow depth. You may expect contribution of cross terms (i.e. vn.vm)

p16, l. 2: "Mode 2 dominates over most of the ETNA except offshore at latitudes beyond $12°N$ where mode 1 dominates." Mode 2 seems to dominate over Mode 1 almost every where?

P16, l. 7: "Dominance of mode 2 is a well understood attribute of equatorial/tropical regions (Philander and Pacanowski, 1980)" This depends on stratification and variability. This appears as an excessive generalization. For instance, this is not the case in the equatorial central Pacific. Please rephrase.

p. 16, l. 10-12: "Using the phase speeds indicated in Fig. 15, we find 22oN, 11oN and 7oN for the critical latitudes associated with vertical mode 1, 2 and 3 respectively, in agreement with previous estimates (Clarke and Shi, 1991)." From Table 1 of Clarke and Shi (1991) the first segment ($\sim 8°$S) is already beyond the critical latitude of the semi-annual frequency for the second baroclinic mode. Please rephrase. In fact is seems like that linear theory does not apply for the semi-annual cycle?

p. 16, l. 15-16: "Over the continental slope, the progressive deepening of the WABC with increasing latitude in Fig. 9 corresponds to a reduction in the contribution of high-order modes". I would say that this is the contrary, the deeper the WABC, the likely larger contribution of the higher-order modes. This statement would need to be either modified or supported by a specific analysis.

p. 16, l. 22-24: "The northern end of this sector coincides with the critical latitude for baroclinic mode 1 semi annual RWs, hence a potential resonant excitation of these waves because their group velocity vanishes (Hagen, 2005)." Not clear. Please clarify what you mean by "potential resonant excitation".

---

## Author Comment (AC1) · 12 Jul 2018

We deeply thank the reviewers for their careful reading and commenting of our manuscript. We have strived to address their concerns and made numerous modifications of the text and figures. Point by point description of the changes (in blue) is provided below following every comment of the reviewer (in black).

Comments by the reviewers on the Rossby wave section have led us to discover an error in the value we were using for the mode 1 deformation radius at 14^oN (50 km instead of 58 km - the gravity phase speed given in Fig 16 was nonetheless correct). This has led to substantial modifications of section 6, figure 8 and figure 14.

Note that without being asked to do so, we have slightly modified our notations and now refer to the depth anomaly for an isotherm, eg., 18oC, as $\delta z_{18}$ and no longer $z_{18}$. (see text p. 6 l. 13-15).

Reviewer 1:

While in general the writing is clear and correct, a rather relaxed and somewhat conversational style leads to lengthy text that could be made more terse, with a modicum of effort. Examples include "are quite noisy but the overall impression is that they tend to be more toward the west. . . p7 l24) or "note also that significant small-scale noise results from the sinuosity of the flow past topographic irregularities whose position can thus locally depart from the shelf break, e.g. in relation to standing meanders in the lee of headlands, as noticeable in . . . p13 l10". Quite so, but it could surely be said more succinctly.

We have made numerous modifications of the text to try to improve its readability.
Specifically, p. 7 l. 24-25, we now say:
"North of 10oN within 5-10o from the West African coast observed zonal velocities are weak and variable but generally oriented westward. This westward tendency is less marked for model velocities."
p. 13 l. 25-28, we now say: "Note also that standing meanders of the WABC past topographic irregularities can produce substantial excursions of the flow away from the shelfbreak, hence rapid alongshore changes of $V_{g}^{26.7}$ (see Fig.10; the position of the main capes is indicated in Fig.12).

Figures are inconsistently labelled, some with panels marked by letter, others not. Even when so labelled, the panels are in some cases referred to not by letter but by position.

We have added labels wherever needed (Figs. 1, 7, 9 et 17), modified captions accordingly, and made sure figures are referred to by these labels throughout the manuscript. Labels for Fig. 8 which were already present are now used in the text.

Abstract: p1 l4 "a suite of" is unnecessary l6 "southernmost" rather than "outmost"
We have modified the text as suggested.

Introduction: p2 l5 preferable to refer to "northwest Africa" for generality and to avoid controversy with respect to the internationally disputed borders in the area.
We have modified the text as suggested.

p2 l5 Delete"see geographical and oceanographical"
We have modified the text as suggested.

p2 l9-10 The water mass front and its weakening with depth have been recognized long before the work cited, e.g. Allain (1970), Fraga (1973) and other papers of that period, and more over later years.
The text now reads (p2 l9-10): "At depths greater than ~300 m, northern Atlantic central waters are found further south and the water mass contrast fades away (e.g., Fraga,1974; Tomczak, 1981; Pena-Izquierdo, 2015).

P4 l19 "literature" not "litterature"
This has been corrected as well as at line 10.

Model evaluation: p7 l16 Figure 6 is called out of order, before Figure 3.
Figures have been reordered such that Fig 6 is now Fig 3.

p8 ll5-12 The doming structure on the zonal line in Fig 2 is weaker in the model. The localized uplift at almost all levels near coast is not mentioned. Incidentally although the figure panels are labelled a-d, the legend refers to left,right, top and bottom.
We agree that the doming structure on the zonal line is weaker in the model. This is why the text mentioned (and still does at p. 8 l. 8) " … albeit with less amplitude than found in the observations." The legend of new Fig. 3 now uses the labels of the panels.

Seasonal cycle of the WABC: p10 l10 State "Alongslope vertical sections of ..." to emphasize the difference with Figure 7. p10 l11 "across-shore averaging" would be better expressed as "across-slope averaging" as the shore bounds the flow. This occurs in a few other places in the text.
We have modified the text and used "slope" in every instance where it was more accurate than "shore" (where indicated by the reviewer - p. 10 l. 21-22 and also p. 7. l. 6,  p13 l. 6 and 8).

WABC coastal dynamics: p11 l10 "as more classically estimated" - does this mean calculated from the observations? If so, please state it more directly.
We meant to refer to the lower bound frequently chosen to compute Sverdrup transport in previous studies. We now say (p. 11. l. 19-20) "as more commonly estimated in past studies" and cite Marchesiello et al, 2003 and Small et al, 2015 (which were already part of the reference list).

P12 l15 Delete "it" before "as occurs".
We have modified the text as suggested.

---

## Author Comment (AC2) · 12 Jul 2018

We deeply thank the reviewers for their careful reading and commenting of our manuscript. We have strived to address their concerns and made numerous modifications of the text and figures. Point by point description of the changes (in blue) is provided below following every comment of the reviewer (in black).

Comments by the reviewers on the Rossby wave section have led us to discover an error in the value we were using for the mode 1 deformation radius at $14^\circ$N (50 km instead of 58 km - the gravity phase speed given in Fig 16 was nonetheless correct). This has led to substantial modifications of section 6, figure 8 and figure 14.

Note that without being asked to do so, we have slightly modified our notations and now refer to the depth anomaly for an isotherm, eg., $18^\circ$C, as $\delta z_{18}$ and no longer $z_{18}$. (see text p. 6 l. 13-15).

Reviewer 2:
This paper investigates the forcing mechanism of the poleward undercurrent at an- nual and semi-annual frequencies, focusing on the region between $10^\circ$N and $20^\circ$N. Four distinct processes are looked at: i) local generation of a poleward undercurrent in conjunction with coastal upwelling conditions ii) remote forcing of poleward flow with subsequent propagation in the form of coastal trapped waves iii) local modulation of the nearshore Sverdrup transport in relation with the seasonal cycle of the wind stress curl iv) Rossby wave modes at the semi-annual frequency. The analysis is based on a forced experiment with an OGCM at the resolution $1/4^\circ$ over the period 1979-2015.

Considering the scarcity of observations in that region, the model simulation is viewed as material for improving our knowledge of the boundary current dynamics in this region at seasonal frequency, recognizing that the model set up has certainly limitations (e.g. resolution) to address all aspects of the variability. The analysis provides interesting insights in the variability in this region. It conveys in particular the idea that the westward propagation of energy away from the coastal guide is an important process contributing to the poleward attenuation of the boundary current. This westward propagation of energy is interpreted as resulting from the propagation of a semi-annual/annual extra- tropical Rossby wave, although the agreement between model and linear theory indicates that non-linear dynamics is certainly involved. The analysis also suggests that the poleward flow in the southern part of the Canary current system has a remote component originating from the Gulf of Guinea (i.e. wind forced), which contrasts with former studies that invoke an equatorial origin.

The analysis is based on the estimate of Sverdrup transport and geostrophic meridional transport within a priori determined depth ranges, so as to discriminate locally wind forced process and remote effects (equatorial origin or along the coastal wave guide). It makes use of the linear theory for the interpretation of the variability.

The paper is well written and provides a nice overview of all the potentially important processes in this region that are tested based on the medium resolution model simulation, offering a benchmark for other model analysis and material for the interpretation of data. While it is useful to have this broad perspective of the variability, the caveat is that it feels sometimes that the paper lacks focus making it difficult to retain the main result. I think this problem can be easily overcome through improvement of the presentation. For instance, the main conclusions could be clarified through providing a schematic summarizing the main processes and highlighting the findings of the paper. It might also be useful to summarize in a table the main processes that have been tested, through which diagnostics, and the consistency with previous studies.

We have slightly changed parts of the Discussion/conclusion.
New pieces include (p. 19 l. 22-24):
"This is an important difference with the annual cycle of the boundary slope current discussed in several past studies including Mittelstaedt (1991) and Lazaro et al (2005). Note though that important signs of a semi-annual cycle can be seen in Lazaro et al (2005) in which two along-slope transport maxima across their so-called section B are found.
The main change is the addition of Fig. 18 called at p. 21 l. 1.

Owing to this remote forcing, largest WABC transports occur while wind stress curl is relatively weak and upwelling winds intensify, i.e., local forcing is least conducive to poleward flow.

They are also in some instances in the text some unclear (or too vague) statements (see specific comments).

We have made a number of adjustments to the text, in particular in response to the reviewer's comments (see below for details; see also changes made in response to comments of reviewer 1).

General comments:

1) The title should reflect that the paper mostly investigates the seasonal variability of the poleward flow. In fact the paper appears to me as a study of the forcing mechanism of the semi-annual variability more than a study of the dynamics of the shadow zone which refers to the mean deep circulation?

The reviewer is correct that the terminology "shadow zone" does not accurately present what is being studied in this manuscript. But the title is made of two pieces, the first one being common to Part1 and Part2. And we firmly believe the reference to "shadow zone" is appropriate to characterise the broader context of the two manuscripts making up this study. The second piece of the title "the poleward slope currents along West Africa" more specifically describes the content of this Part1 manuscript.

The title of Part2 on which we are actively working will be "A model perspective on the dynamics of the shadow zone of the eastern tropical North Atlantic. Part 2: sources of coastal upwelling waters and regional circulation features". This study will be concerned with the circulation over the same depth range as Part1 but considered at larger spatial scale. It will focus on the connections between the WABC and the open ocean, on the significance of the Guinea dome for the regional circulation, on the processes involved in the formation of the peculiar PV structure presented in Part1, and on the significance of the multiple zonal jets present in the ETNA. We see these two papers as a coherent work giving account of the weak but finite circulation of the ETNA shadow zone, hence the title we chose and would rather keep.

2) The section 6 is devoted to the analysis of the meridional flow seasonal variability in terms of extra-tropical Rossby wave. On the one hand the authors suggest that a semi- annual Rossby wave can radiate off-shore at latitudes South of ~ 15°N but in Section 5 they also show that there is a possible remote source of semi-annual poleward flow off WA, implying the propagation of Coastal Trapped Waves at the semi-annual frequency (forced by the winds along the coast of Ivory coast). In the frame of linear theory, at a given frequency, a wave is either trapped along the coast or radiates off-shore, so could you try to reconcile these apparent conflicting results (or clarify the text). This calls also to clarifying what is the forcing mechanism of the semi-annual Rossby wave that is discussed.

The generation of westward propagating Rossby waves by coastal trapped waves has been the subject of a few papers in the past decades. Specifically, Kelvin waves or CTWs are exact solutions in the f-plane approximation only, eg, for the shallow-water 1 1/2 layer equation set. On a beta-plane (or in the real ocean) some scattering occurs for the low-frequency waves because the underlying fluid motion is sensitive to the meridional gradient of the Coriolis parameter.

The text (p. 16, l. 5-8) has been slightly modified to better introduce the process at play and refer to the key studies on this subject: "… Figs. 7, 8 and 10 are consistent with the generation of semi-annual Rossby waves at the WA eastern boundary via the scattering of coastal waves due to the meridional change of the Coriolis parameter (McCalpin 1995; Ramos et al, 2006).

Other comments:

p. 2, l. 28-29: "in part because the shape of the African continent produces a curvature of the trade winds" through which process?

We now explicitly mention "sea-land contrasts" to make sure the reader follows our argument. The text now reads (p. 2 l. 28-29): " … because sea-land contrasts and the shape of the African continent produce a curvature of the trade winds favorable to cyclonic rotation, and also because the ETNA is a transition region toward the ITCZ (\ie the trade wind intensity gradually drops southward).

p.3, l. 4: AWA not defined
The AWA research program is now ended and no follow-up program has yet formally emerged. We now say things more loosely (p. 3 l. 4): "As part of a research effort aimed at implementing an ecosystem approach to manage the WA marine environment and fisheries, we are concerned …"

p. 9, l.4-5: Not clear here if the amplitude the semi-annual and annual cycles was actually estimated in observations and model. How do you estimate the 50%?. It would be useful to indicate an error bar (dispersion) associated to the number of years that is retained to calculate the mean of Fig. 5 considering that the observations of Picaut corresponds to the period 57-64. You could select randomly chunks of 8 years in the model and provide the dispersion among the ensemble chunks. This would inform also on the possible influence of decadal variability on the validation.
In Fig. 5 (now Fig. 6), we are now showing an "error bar" estimated from the minimum and maximum depth for the position of 4 isotherms (11, 15, 18 and 21oC) when 8-year climatologies are constructed. The caption of Fig.6 has been modified accordingly.
We have also modified the text to clarify the quantitative assertion (p. 9 l. 5-6): "On the other hand, a quantitative difference concerns the amplitude of the oscillations which are underestimated by 50% or more in the model, eg, 30 m (resp. 50 m) peak to peak amplitude for the seasonal cycle $\delta z_{16}$ in TROP025 (resp. in the observations).

p. 9, l. 7-9: "To put these biases into perspective it would be useful to know the degree to which the observations in Picaut (1983) are representative of a long alongshore stretch of ocean, as opposed to very local conditions that TROP025 cannot represent due to lack of resolution." Not clear. Do you mean that the observations of Picaut (1983) would account for coastal trapped wave variability, which may not be well simulated in TROP025 owing to the too coarse resolution? What is the critical latitude of the annual and semi-annual period? It could be useful to mention this information at that stage for clarity.
Our formulation was confusing and we have clarified with a modification of the text which now reads (p. 9 l. 10):"To put these biases into perspective it would be useful to know the degree to which very local conditions at 4$^o$N, 5$^o$W that TROP025 does not represent (\eg fine-scale irregularities of the shoreline or continental shelf/slope bathymetry) contribute to the observed seasonal cycle." Presenting critical latitudes at this point makes the text more complicated and we have refrained from doing so. We believe our argument can be made without it.

Figure 8: indicate latitude ranges? We assume it is the same than Figure 7?
We forgot to provide this information, which is indeed the same than for Figure 7. The caption has been corrected and now indicates : "… averaged alongshore between 13$^o$ and 15$^o$ N."

p. 9, l. 28-30: "In winter and summer a core of poleward flow present a few hundred kilometers from shore is suggestive of westward propagation of the poleward undercurrents." This is not straightforward. It is more like there is an annual Rossby wave that is in phase with the seasonal cycle of the undercurrent? The concept of an undercurrent propagating off-shore as a Rossby wave is not straightforward since the undercurrent is usually tight to the bottom (slope) boundary layer dynamics. The text may need to be clarified.
We have clarified and the text now reads (p. 9 l. 32) :"In winter and summer a core of poleward flow present a few hundred kilometers from shore is suggestive of the radiation of westward propagating Rossby waves from the continental slope, as also found in other regions (Vega et al, 2003; Ramos et al, 2006; Colas et al, 2008; Rao et al, 2010), particularly in the tropics.

p. 10, l. 5-6 "Zonal integration is performed from the coast to the first offshore location where the flow changes direction, so the width over which this transport is achieved varies in time." Is it possible to plot the distance from the coast over which the zonal integration is performed on Figure 8a.
We have added this information in Fig. 8a and modified the caption accordingly. Doing so has revealed a small bug in the program computing transport (Fig 8b) which led to reporting an erroneous value for the month of october only. Examination of the new figure Fig.8b has led us to

modify the final paragraph of section 4 to emphasise the manifest asymmetry between the two phases Ps and Pf.

We now say (p. 11 l. 13):"In contrast, the winter time interval from Pf to Ps appears to be a bit shorter than the summer interval from Ps to Pf (e.g., poleward transport is present nearshore in Feb. but still absent in Aug.). This is confirmed at 14oN by inspection of Fig. 8a. Asymmetry between the meridional flow during Ps and Pf is more generally confirmed by Fig. 8b which reveals a sharper peak of northward transport for the latter period."

Note that we now show a Hovmoller plot also at 8^oN (Fig. 8 panel c). The calculation of the transport at that latitude has led us to impose a 3^o limit to the size of the across-shore integration sector. This is specified in the caption and text (p. 10 l. 14).

p. 10, l. 27-31: You could do maps of the annual (semi-annual) amplitude and phase of the meridional transport of Figure 10 in order to support the interpretation and provide more quantitative statement.

In the course of the study, we have used harmonic analysis to compare the relative importance of the annual versus semi-annual periods. We reproduce one of them below for the ratio of the amplitudes of the semi-annual to annual harmonics for v (at depth 100 m). It clearly shows the dominance of the semi-annual harmonics.

Because we already have 18 figures, we decided against adding one more. On the other hand, we have added a footnote to describe the outcome of this analysis (footnote 2 in page 11).

p.11, l. 30.31: It is not clear why it should be "resonant excitation of free Rossby wave modes". It could be locally forced Rossby waves or remotely forced Rossby waves? Can we have purely "resonant" extra-tropical Rossby waves at these latitudes without coupling with wind stress? Any reference to support such a hypothesis considering that the references that are provided are for resonant modes in the equatorial wave guide?

The reviewer correctly points an error present in the text. It is simply "local excitation of free Rossby wave modes" that should be mentioned here. "resonant" has been removed from the text (p. 12 l. 18). Note that remotely forced Rossby waves are not appropriate because we are concerned with nearshore transport and Rossby waves will only affect the area near their forcing region and further west.

p.14, l. 20-21: "With respect to 4), z18 and V26.7,g are not precisely in phase as they are expected to be for theoretical Kelvin waves (Cushman-Roisin and Beckers, 2011)". This assumes a 1.5-layer (i.e. one barocline mode) dynamics. In a multimode context you can have a phase difference.

Agreed. We now say (p. 15 l. 9-10)" … as they are expected to be for theoretical Kelvin waves in a model for a single baroclinic mode (e.g., reduced gravity model; Cushman-Roisin and Beckers, 2011)."

p. 14, l. 24 :" Due to Rossby waves pressure fluctuations associated with CTWs propagate offshore" This is not clear. Please rephrase.

We have reformulated this sentence to clarify (p. 15 l, 13-14): "Radiation of Rossby wave from the coastal wave guide implies that pressure disturbances associated with CTW dynamics propagate offshore. "

p.15, l. 15: ". . .the presence of a semi-annual Rossby wave coupled to the coastal trapped wave activity". This is unclear. Do you mean the coastal trapped wave activity is concomitant with the semi-annual Rossby wave. From a theoretical point of view, you cannot have a Rossby wave and a coastal trapped wave at the same frequency. The wave is either trapped or radiate as Rossby wave.

We believe this formulation is ok. Kelvin waves or CTWs are only exact solutions in the f-plane approximation. As mentioned above, some dispersion of the low-frequency waves occurs on a beta-plane which produces this situation where the coastal wave propagates poleward but also leaks energy westward. This is perhaps most easily understood with the formalism developed in McCalpin 1995.
The text page has been slightly modified to mention the process at play and refer to the key studies on this subject (p. 16 l. 5-8): "… Figs. 7, 8 and 10 are consistent with the generation of semi-annual Rossby waves at the WA eastern boundary via the scattering of coastal waves due to the meridional gradient of the Coriolis parameter (McCalpin 1995; Ramos et al, 2006).

p.15, l. 20: "Rn= 50 km which is approximately the value we find at 14∘N,18,30∘W for the first deformation radius, based on the model stratification". How do you calculate Cn in the model? More details on the method used to derive the baroclinic mode structure should be provided. In particular, do you use "continuous" (interpolated) profiles or does the modes are derived on the model vertical grid?

We have added some details on the description of the method and the text in this section has been substantially modified. At p. 16 l.21-23 we now say:
"In what follows baroclinic mode characteristics are determined based on the TROP025 stratification computed on the model 75 grid levels. The calculation is made with the dynmode program available at https://woodshole.er.usgs.gov/operations/sea- mat/klinck-html/index.html.".

p. 16, l. 1.: "integration to a layer in which the poleward flow is concentrated, e.g., the upper 200 m, leads to similar results". This is surprising since the baroclinic modes are no longer orthogonal when integrated over such a shallow depth. You may expect contribution of cross terms (i.e. vn.vm)

We think this lack of sensitivity is not directly related to the orthogonality of the modes because vertical integration of kinetic energy over the entire water column or over the upper 200 m is a final arbitrary diagnostic aimed at quantifying the relative importance of the different modes once decomposition has been performed. Our result merely shows that irrespective of the norm that is being chosen (full water column kinetic energy or upper ocean kinetic energy) mode 2 dominates.
We have slightly changed the wording to draw the reader's attention on the meaning of this sentence (p. 17 l. 7).
"Restricting the final step of vertical integration to a layer in which the poleward flow is …".

p16, l. 2: "Mode 2 dominates over most of the ETNA except offshore at latitudes beyond 12oN where mode 1 dominates." Mode 2 seems to dominate over Mode 1 almost every where?

We were trying to phrase the argument in a careful way but we overdid it. We agree with the reviewer. We now say (p. 17 l. 10) "Mode 2 dominates over most of the ETNA except for a few isolated offshore grid cells in the latitude range 10-15oN where mode 1 is of comparable amplitude".

p16, l. 7: "Dominance of mode 2 is a well understood attribute of equatorial/tropical regions (Philander and Pacanowski, 1980)" This depends on stratification and variability. This appears as an excessive generalization. For instance, this is not the case in the equatorial central Pacific. Please rephrase.
We agree and now say (p. 17 l. 9-10) "Dominance of mode 2 is a well understood attribute of *many* equatorial/tropical regions …".

p. 16, l. 10-12: "Using the phase speeds indicated in Fig. 15, we find 22oN, 11oN and 7oN for the critical latitudes associated with vertical mode 1, 2 and 3 respectively, in agreement with previous estimates (Clarke and Shi, 1991)." From Table 1 of Clarke and Shi (1991) the first segment (~8oS) is already beyond the critical latitude of the semi-annual frequency for the second baroclinic mode. Please rephrase. In fact is seems like that linear theory does not apply for the semi-annual cycle?
There seems to be some misunderstanding somewhere. Table 1 of Clarke and Shi (1991) for NE Atlantic (mode 1) has critical periods about 180 days for coastal segments 8 (184.6 days) and beyond. Referring to their Fig2, we find coastal segment 8 to be situated just above Cape Blanc, 21oN, hence our comment on the fact that our results agree with those of Clarke and Shi (1991) (for higher modes, rescaling first mode phase speed also gives good agreement). Presentation of critical latitudes (now p. 16 l. 17-26) has been clarified as part of the complete overhaul of section 6.

p. 16, l. 15-16: "Over the continental slope, the progressive deepening of the WABC with increasing latitude in Fig. 9 corresponds to a reduction in the contribution of high- order modes". I would say that this is the contrary, the deeper the WABC, the likely larger contribution of the higher-order modes. This statement would need to be either modified or supported by a specific analysis.
Mode 2 and 3 have a vertical structure whose sign at depth below 250 m is opposite to that for the surface (this is true even closer to the surface for mode 3). Hence, some cancellation between near-surface and deep contribution will reduce the projection on mode 2 and 3 where the WABC reaches below 250m. We have slightly expanded the text which now reads (p. 17 l. 21-23):
"Over the continental slope, the progressive deepening of the WABC with increasing latitude (Fig. 9) corresponds to a reduction (resp. increase) in the relative contribution of high-order (resp. low) modes, \eg less weight on mode 3 whose upper zero-crossing is at 75 m (Fig. 16).

p. 16, l. 22-24: "The northern end of this sector coincides with the critical latitude for baroclinic mode 1 semi annual RWs, hence a potential resonant excitation of these waves because their group velocity vanishes (Hagen, 2005)." Not clear. Please clarify what you mean by "potential resonant excitation".
We meant that there is the possibility of a resonant excitation of mode 1 semi-annual RWs because they are associated with vanishing group velocity. The text has been clarified (p. 17 l. 30-31): " … hence the possibility of resonant excitation of these waves because their group velocity vanishes [Hagen 2005].